# Single-nucleus multiomics reveals the gene regulatory networks underlying sex determination of murine primordial germ cells

Adriana K Alexander[1], Karina F Rodriguez[1], Yu-Ying Chen[1], Ciro Amato[1], Martin A Estermann[1], Barbara Nicol[1], Xin Xu[2], Humphrey HC Yao[1]*

[1]Reproductive Developmental Biology Group, National Institute of Environmental Health Sciences, Research Triangle Park, Durham, United States; [2]Epigenetics & Stem Cell Biology Laboratory, National Institute of Environmental Health Sciences, Research Triangle Park, Durham, United States

## eLife Assessment

This **important** study reports single-nucleus multiomics-based profiling of transcriptome and chromatin accessibility of mouse XX and XY primordial germ cells (PGCs). The main conclusions of this study, which will be of interest to developmental and reproductive biologists, as well as andrologists, are supported by **convincing** data.

*For correspondence:
humphrey.yao@nih.gov

Competing interest: The authors declare that no competing interests exist.

**Abstract** Accurate specification of female and male germ cells during embryonic development is critical for sexual reproduction. Primordial germ cells (PGCs) are the bipotential precursors of mature gametes that commit to an oogenic or spermatogenic fate in response to sex-determining cues from the fetal gonad. The critical processes required for PGCs to integrate and respond to signals from the somatic environment in gonads are not well understood. In this study, we developed the first single-nucleus multiomics map of chromatin accessibility and gene expression during murine PGC development in both XX and XY embryos. Profiling of cell-type-specific transcriptomes and regions of open chromatin from the same cell captured the molecular signatures and gene networks underlying PGC sex determination. Joint RNA and ATAC data for single PGCs resolved previously unreported PGC subpopulations and cataloged a multimodal reference atlas of differentiating PGC clusters. We discovered that regulatory element accessibility precedes gene expression during PGC development, suggesting that changes in chromatin accessibility may prime PGC lineage commitment prior to differentiation. Similarly, we found that sexual dimorphism in chromatin accessibility and gene expression increased temporally in PGCs. Combining single-nucleus sequencing data, we computationally mapped the cohort of transcription factors that regulate the expression of sexually dimorphic genes in PGCs. For example, the gene regulatory networks of XX PGCs are enriched for the transcription factors, TFAP2c, TCFL5, GATA2, MGA, NR6A1, TBX4, and ZFX. Sex-specific enrichment of the forkhead-box and POU6 families of transcription factors was also observed in XY PGCs. Finally, we determined the temporal expression patterns of WNT, BMP, and RA signaling during PGC sex determination, and our discovery analyses identified potentially new cell communication pathways between supporting cells and PGCs. Our results illustrate the diversity of factors involved in programming PGCs toward a sex-specific fate.

## Introduction

Proper formation of germ cells during embryonic development is crucial for ensuring the production of functional gametes (*McLaren, 2003*). The embryonic precursors to mature gametes, primordial germ cells (PGCs), are the bipotential stem cells of the germline that give rise to eggs and sperm (*McLaren, 2003*; *Hancock et al., 2021*). During embryonic development in mice, PGCs commit to the oogenic or spermatogenic lineage in response to sex-determining cues from the gonadal environment in a process termed PGC sex determination (*Spiller and Bowles, 2015*). During sex determination, XX PGCs in the fetal ovary commit to oogenesis by entering meiosis immediately after pre-granulosa cell specification (*Borum, 1961*). By contrast, XY PGCs in the fetal testis initiate the spermatogenic program and arrest mitotically in response to signals from Sertoli cells (*Spiller and Bowles, 2015*; *Bowles and Koopman, 2010*). Defects in germ cell differentiation, particularly during fetal life, often lead to reproductive diseases, such as infertility (*Czukiewska and Chuva de Sousa Lopes, 2022*) and the formation of germ cell tumors (*Oosterhuis and Looijenga, 2019*). Thus, it is necessary to enhance our understanding of germ cell development so we can better identify the etiologies of reproductive dysfunction in humans.

PGC sex determination is induced by the sex-specific activation of transcription factors (TFs) and downstream gene networks (*Spiller and Bowles, 2022*). In XX PGCs, the retinoic acid (RA)-responsive TFs STRA8 and MEIOSIN and the bone morphogenetic protein (BMP)-responsive TF ZGLP1 are required for entry into meiosis and oogenesis (*Spiller and Bowles, 2022*; *Ishiguro et al., 2020*; *Nagaoka et al., 2020*). These TFs also initiate the expression of genes related to meiotic processes, including *Rec8* and *Sycp1-3* (*Spiller and Bowles, 2022*). In contrast, XY PGCs require the expression of cell cycle inhibitors, such as *Bnc2* and *Cdkn2b*, and the male-specific genes, *Nanos2* and *Dnd1*, to enter mitotic arrest (*Vanhoutteghem et al., 2014*; *Spiller et al., 2010*; *Saba et al., 2014*; *Cook et al., 2011*). During the transition from PGC to oogonium or gonocyte, both XX and XY PGCs must also lose their bipotential state by downregulating the pluripotency-related genes, *Pou5f1, Nanog,* and *Sox2* (*Spiller and Bowles, 2022*). Consequently, there are multiple layers of gene regulation required for sex determination of PGCs.

Beyond the patterns of gene expression in PGCs, relatively little is known about how signals from the gonadal environment activate the expression of sexually dimorphic TFs and genes in PGCs. First, the gene regulatory networks, *i.e.*, TFs and their predicted target genes, specific to XX and XY PGCs are not well defined. Second, it remains unclear how the chromatin environment is temporally regulated during PGC sex determination. Finally, additional data are needed on the patterns of ligand-receptor expression in gonadal supporting cells and PGCs. Previous reports have used bulk gene regulation and expression genomics assays to investigate the transcriptional programs underlying PGC development (*Houmard et al., 2009*; *Jameson et al., 2012*; *Lesch et al., 2013*; *Rolland et al., 2011*). However, these assays may not have the resolution or sensitivity required to detect the transient changes in gene regulation among PGC subpopulations that are essential to sex determination.

In the present study, we employed the integrative genomics method, combined single-nucleus transcriptome and chromatin accessibility sequencing from the same cell, to decipher various layers of gene regulation during PGC sex determination in mice. We comprehensively profiled 3,054 XX and XY PGCs at embryonic days (E) E11.5, E12.5, and E13.5, which covers the developmental time frame from bipotential to sexually differentiated PGCs. Single-nucleus sequencing enabled the detection of sex-enriched regulatory loci, TFs, and gonadal cues that may be responsible for initiating gene expression in individual PGCs. We first systematically examined the molecular signatures of PGC subpopulations to identify the genes and accessible chromatin regions underpinning the sex-specific fates of PGCs. By combining our epigenomic and transcriptomic data, we predicted *cis*-regulatory elements and sex-enriched TFs to construct the gene regulatory networks unique to XX and XY PGCs. Lastly, we probed the cell-cell communication pathways between supporting cells and PGCs to nominate potentially new ligand-receptor pairs involved in PGC development. Our results provide insights into the cell fate decisions underlying gametogenesis and sex determination of PGCs.

## Results

### Combined multiome analysis of single-nucleus gene expression and chromatin accessibility of mouse PGCs

To understand how intrinsic transcription networks inside PGCs and external cues from the gonadal environment control PGC sex determination, we performed combined snRNA-seq and snATAC-seq (10x Genomics Multiome) for all cell populations in mouse fetal gonads. This approach allowed us to obtain paired gene expression and chromatin status from the same cell for all cell populations from female (XX) and male (XY) gonads. Whole gonads from Nr5a1-cre x Rosa-tdTomato9 embryos were collected at embryonic days (E) E11.5, E12.5, and E13.5, which encompass the developmental time frame of sex determination: initiated (E11.5), sexually differentiating (E12.5), and morphologically dimorphic (E13.5) gonads. Sequencing libraries of gonads for each sex and embryonic day were generated from two independent technical replicates for an average of 8,687 nuclei/sample with a sequencing depth of ~75,544 reads/nucleus and ~5755 UMIs/nucleus. Based on the well-established germ cell markers *Ddx4* (*van den Bergen et al., 2009*) and *Pou5f1*(*van den Bergen et al., 2009*), we identified 3,054 germ cells. Somatic supporting cell populations were identified using *Runx1* expression as a marker of pre-granulosa cells in our female datasets and *Sox9* expression as a marker of Sertoli cells in our male datasets (*Stévant et al., 2019*). A combined 22,382 XX and XY supporting cells were identified. Together, these data allowed us to determine the clustering patterns of differentiating PGCs, identify regulatory loci and TFs enriched in PGCs, and find previously unreported interactions between PGCs and gonadal supporting cells (*Figure 1a*).

We first asked whether single-nucleus gene expression and chromatin accessibility information can quantify the similarity or differences within PGC populations. We clustered single PGC nuclei in low-dimensional space based on their snRNA-seq (*Figure 1b–d*) or snATAC-seq (*Figure 1e–g*) data using Uniform Manifold Approximation and Projection (UMAP) for dimensional reduction. This graph-based clustering algorithm unbiasedly separated and clustered individual cells with similar transcriptomic profiles (*Becht et al., 2019*). Such analyses revealed that the transcriptomes of E11.5 XX and XY PGCs overlap extensively, whereas E12.5 and E13.5 PGCs principally separated according to their sex (*Figure 1b*). In addition, we uncovered eight broader clusters of PGCs at a resolution that captured PGC subpopulations (*Figure 1c*). We determined the cell-type composition of the snRNA-seq transcriptome-based clusters by plotting the percentage of PGCs at each embryonic stage and sex (*x*-axis) in each of the eight transcriptome-based clusters (*y*-axis) (*Figure 1d*). First, E11.5 and E12.5 XX PGCs primarily consisted of clusters 0, 1, and 4 (*Figure 1c and d*). At E13.5, XX PGCs separated into three main clusters, 4, 5, and 6, and had significantly higher expression of the synaptonemal complex component and meiotic marker *Sycp1* (*Gray and Cohen, 2016*) and the chromatin modifier *Hdac9* (*Yang et al., 2021*; *Figure 1c and d*; *Appendix 1—figure 2*). In XY gonads, PGCs at E11.5 grouped together into three clusters, 0, 1, and 7 (*Figure 1d*). At E12.5-E13.5, XY PGCs converged onto a single distinct population (cluster 7), indicating less transcriptional diversity among E12.5-E13.5 XY PGCs when compared to E12.5-E13.5 XX PGCs (*Figure 1d*). Furthermore, we found that E12.5-E13.5 XY PGCs clustered separately from other PGC populations due to elevated expression of *Bnc2*, which is required for mitotic arrest in PGCs (*Vanhoutteghem et al., 2014*), and the forkhead-box TF *Foxp1* (*Herman et al., 2021*; *Appendix 1—figure 2*). Finally, a small proportion of all PGC populations (E11.5-E13.5 XX and XY PGCs) were detected in clusters 2 and 3, and displayed enriched expression of ribosome biogenesis genes and genes with reported roles in reproduction, such as the mitochondrial leucyl-tRNA synthetase *Lars2* (*Pierce et al., 2013*) and the DNA repair protein *Mgmt* (*da S Martinelli et al., 2017*; *Figure 1d*; *Appendix 1—figure 2*). Together, these data indicate that the transcriptomic profiles of PGCs are largely homogenous within each sex at E11.5. Subsequently, XX PGCs maintain their transcriptional heterogeneity at E12.5-E13.5, whereas XY PGCs transcriptionally converge onto a single identity beginning at E12.5.

The snRNA-seq clustering revealed transcriptomic characteristics of PGCs. Next, we performed unbiased clustering of PGCs according to their aggregate profiles of snATAC-seq peaks (210,666 total peaks) to detect the patterns of chromatin accessibility associated with PGC development (*Figure 1e and f*). The snATAC-seq-based clustering of PGCs showed that the chromatin status of E11.5 XX and XY PGCs was highly similar (*Figure 1e*). By contrast, E12.5-E13.5 XX and XY PGCs formed distinct groupings at opposite poles of the UMAP (*Figure 1e*). Unbiased graph-based clustering of PGCs uncovered nine clusters of PGCs with unique stage- and sex-specific snATAC-seq



**Figure 1.** Single-nucleus multiomics sequencing of E11.5-E13.5 XX and XY primordial germ cells (PGCs). (**a**) Workflow of single-nucleus multiomics analyses. (**b–c**) Dimensional reduction (Uniform Manifold Approximation and Projection [UMAP]) and cell-type-specific clustering of snRNA-seq data collected from E11.5-E13.5 XX and XY PGCs. Each dot represents an individual cell color coded by either embryonic stage and sex (**b**) or cluster number (**c**). (**d**) Cell-type composition of snRNA-seq clusters. The embryonic stage and sex of the PGC population is indicated on the *x*-axis and the

*Figure 1 continued on next page*

*Figure 1 continued*

unbiased clustering number is represented on the *y*-axis. (**e–f**) UMAP of snATAC-seq data collected from E11.5-E13.5 XX and XY PGCs color coded by either embryonic stage and sex (**e**) or cluster number (**f**). (**g**) Cell-type composition of snATAC-seq clusters. (**h–i**) UMAP visualization of E11.5-E13.5 XX and XY PGCs using a joint, or weighted, measurement of snRNA- and snATAC-seq modalities. (**h**) Trajectory of transcriptomic changes (also known as pseudotime analysis in Monocle) among PGCs overlaid on the joint UMAP. Arrows indicate the path of the developmental trajectory. Numbers on the pseudotime trajectory indicate the following: '0' represents the starting cell population; '1' indicates the branching point in the trajectory path; and '2' represents the terminal cell populations of the trajectory paths. Each PGC is color coded by embryonic stage and sex. (**i**) Unbiased clustering of PGCs using combined snRNA- and snATAC-seq data. Each PGC is color coded by cluster number. (**j**) Cell-type composition of joint UMAP clusters. The number of PGCs per sex and embryonic stage are: 375 E11.5 XX PGCs; 1106 E12.5 XX PGCs; 750 E13.5 XX PGCs; 110 E11.5 XY PGCs; 465 E12.5 XY PGCs; and 348 E13.5 XY PGCs.

peak profiles (*Figure 1f*). First, E11.5-E12.5 XX and XY PGCs together predominantly consisted of accessibility-based clusters 0, 1, and 2, indicating that these germ cells share a highly similar chromatin profile prior to differentiation (*Figure 1f and g*). Second, E13.5 XX PGCs were separated into clusters 3, 4, and 5 (*Figure 1f and g*), which may be due to differences in meiotic entry or progression within these populations of germ cells. Third, E12.5-E13.5 XY PGCs comprised of snATAC-seq clusters 6 and 7, suggesting that XY PGCs may adopt a sex-specific chromatin status beginning at E12.5 (*Figure 1f and g*). Finally, cluster 8, which contained PGCs from both sexes at each stage, was separated from all other accessibility-based clusters (*Figure 1f and g*). Taken together, our findings indicate that the chromatin accessibility profiles of E11.5 PGCs and several E12.5 PGC subpopulations do not show a sex bias, whereas E13.5 XX and XY PGCs clustered by sex.

The multiome datasets provided us the opportunity to define PGC identities based on a more biologically relevant status at the combined transcriptomic and chromatin accessibility levels. Specifically, we integrated gene expression and chromatin accessibility data within a single PGC to obtain a joint, or multimodal, definition of PGC identity (*Figure 1h and i*). Computational integration methods allowed us to co-visualize the variation in both the snRNA-seq and snATAC-seq profiles of PGCs on a single UMAP (*Stuart et al., 2019*). Simultaneously, profiling cell-type-specific transcriptomes and regions of open chromatin for single PGCs resolved the sex-specific trajectories of PGCs with much greater resolution (*Figure 1h and i*). In fact, we observed a strong demarcation between XX and XY PGCs beginning at E12.5 with very little to no overlap between XX and XY PGCs at E13.5 (*Figure 1h–j*). Given that we detected the clearest separation of PGCs when clustering with multimodal data, we analyzed the trajectory of transcriptomic changes (also known as pseudotime analysis in Monocle [*Cao et al., 2019*]) among PGCs overlaid on the joint UMAP (*Figure 1h*). With E11.5 XX and XY PGCs chosen as the starting point (0) of the trajectory analysis, we observed a distinct branching or 'decision' point (1) between XX and XY PGCs at E12.5 (*Figure 1h*). Additionally, our unbiased analysis identified two terminal states (2) of PGCs at the E13.5 XX PGC cluster and E13.5 XY PGC cluster, providing additional confirmation that our data accurately reflect the status of PGC sex determination (*Figure 1h*). We also observed a closed, or circular, branch of the trajectory path among E12.5 XX PGCs (*Figure 1h*), which may arise from the heterogeneous transcriptomes of female PGCs entering meiosis in an asynchronous manner (*Soygur et al., 2021*). These data demonstrate that our approaches enable us to not only visualize the developmental trajectories of PGCs that are clustered with multimodal data, but also empirically trace their sex-specific fates based on changes in gene regulation.

Clustering of PGCs using combined snRNA- and snATAC-seq data identified 11 distinct groups of PGCs and yielded enhanced cluster resolution of PGC subpopulations (*Figure 1i*). This analysis resulted in clusters of PGCs that were predominantly sex-specific and enabled the detection of subpopulations of each PGC type (*Figure 1i*). To better understand the population structure of individual clusters, we plotted the percentage of each PGC type (*x*-axis) for each cluster identified in *Figure 1i* (*y*-axis) (*Figure 1j*). First, E11.5 XX PGCs clustered into one main population, cluster 1, and later diverged into six subpopulations, clusters 0 through 5 at E12.5 (*Figure 1i and j*). Clusters 0–2 were more similar to E11.5 PGCs, whereas clusters 3–5 were composed of E12.5 XX PGCs that were developmentally closer to E13.5 XX PGCs (*Figure 1i and j*). Finally, we found that E13.5 XX PGCs converged onto either one of two developmental fates in clusters 6 and 7, with cluster 7 exhibiting a large spread along the UMAP2 axis (*Figure 1i and j*). These findings indicate that E12.5-E13.5 XX PGCs may require extensive chromatin remodeling and transcriptional

reprogramming to adopt the oogenic fate. By comparison, XY PGCs grouped into clusters 0 and 1 at E11.5 and appeared to converge on a single developmental fate at later stages, with E12.5 XY PGCs comprising clusters 8 and 9 and E13.5 XY PGCs belonging to cluster 10 (*Figure 1i and j*). Given that neither expression- nor accessibility-based clustering alone resolved the granularity of PGC populations to the level described here, these data underscore the empirical power of simultaneously measuring multiple modalities from the same cell to define the developmental fate of PGCs.

## Molecular characterization of genes known for their roles in XY PGC differentiation

We next asked whether our multiomics approaches could gain insight into the regulatory status of genes that are known to be essential for PGC sex determination. We first investigated the patterns of gene expression and chromatin accessibility of male functional genes, *i.e.*, those that have been shown to be functionally important for XY PGC development (*Figure 2*). We found that genes involved in XY PGC development (i.e. *Rb1*, *Rbl2*, *Cdkn1b*, *Cdkn2b*, *Bnc2*, *Cnot1*, *Dnd1*, and *Nanos2*) (*Vanhoutteghem et al., 2014*; *Spiller et al., 2010*; *Saba et al., 2014*; *Cook et al., 2011*; *Suzuki et al., 2014*) showed increasing levels of expression in XY PGCs from E11.5 to E13.5, whereas the expression levels of *Cnot1* and *Nanos3* peaked at E12.5 and E11.5, respectively (*Figure 2a*). Notable XY enrichment of *Bnc2* expression and moderately elevated expression of *Rb1*, *Rbl1*, *Cdkn1b*, *Cdkn2b*, *Nanos2*, and *Nanos3* were observed in XY PGCs compared to XX PGCs (*Figure 2a*). However, expression of *Cnot1* and *Dnd1* showed an XX biased expression pattern (*Figure 2a*). Overall, these data indicate that the majority of male functional genes show a modest increase in expression levels in XY PGCs over XX PGCs. Among these genes, *Bnc2* expression may serve as a definitive marker of XY PGCs at E13.5.

To identify potential mechanisms regulating the expression of XY PGC functional genes, we searched for candidate TFs with XY-enriched gene expression. We first linked snATAC-seq peaks to XY PGC functional genes (listed previously), hereafter termed peak-to-gene linkages, in XY PGCs. Peak-to-gene linkages were determined using Signac functionalities and were derived from the correlation between peak accessibility and the intensity of gene expression. Next, we performed a TF motif enrichment analysis on significant peak-to-gene linkages to identify potential TFs. Finally, we compared the expression levels of potential TFs between XX and XY PGCs to identify TFs with XY-enriched gene expression. This strategy uncovered the Krüppel-like zinc finger family member ZKSCAN5 (zinc finger with KRAB and SCAN domains 5) as a potential regulator of XY PGC functional genes (*Figure 2a*). *Zkscan5* showed XY biased expression (*e.g.*, there is a 1.5-fold increase in the expression of *Zkscan5* between XY and XX PGCs at E13.5), and the level of *Zkscan5* expression increased from E11.5 to E13.5 in XY PGCs (*Figure 2a*). In addition, the ZKSCAN5 motif was not enriched in XX PGCs. ZKSCAN5 is not only highly expressed in the adult testis (*Dreyer et al., 1999*) but is also proposed to regulate cellular proliferation and growth (*Li et al., 2022*). Thus, although little is known about the function of ZKSCAN5, we hypothesize ZKSCAN5 may possess a regulatory role during male germ cell development. These data highlight the strength of single-nucleus multiomics in discovering candidate factors that may regulate the expression of a cohort of developmental genes and, consequently, direct cell fate decisions.

Among the functionally important PGC genes, *Bnc2* expression was most strongly enriched in XY PGCs. We therefore investigated the regulation of *Bnc2* at the chromatin level as an example to demonstrate the utility of combined multiome datasets. The chromatin accessibility of the *Bnc2* promoter was highest in XY PGCs and increased temporally from E11.5 to E13.5 (*Figure 2b*). Furthermore, the level of chromatin accessibility at the *Bnc2* locus was positively correlated with *Bnc2* expression in XY PGCs (*Figure 2b and c*). We then applied the Cistrome Data Browser (Cistrome DB) (*Mei et al., 2017*) to identify putative regulators of *Bnc2* expression based on TF binding motifs detected within 10 kb of the *Bnc2* promoter (*Figure 2d*). The top candidate regulators of the *Bnc2* locus were determined according to a regulatory potential score derived from a combination of TF binding motif information and previously published genomics data (*i.e.*, ChIP-seq, DNase-seq, and ATAC-seq) (*Figure 2d*). Of the top candidate regulators from the Cistrome DB analysis, expression of *Ctcf*, *Nrf1*, *Polr2a*, and *Brd4* was enriched in XY PGCs at E13.5 (*Figure 2e*). In particular, BRD4 is known to activate transcription elongation by recruiting kinases to the largest RNA polymerase II (Pol II) subunit, POLR2A, thereby releasing Pol II from its promoter-proximal bound state (*Jang et al., 2005*). Taken



**Figure 2.** Molecular characterization of genes known for their roles in male primordial germ cells (PGC) differentiation. (**a**) Dotplot of the average expression and percentage of cells expressing *Rb1, Rbl2, Cdkn1b, Cdkn2b, Bnc2, Cnot1, Dnd1, Nanos2, Nanos3,* and *Zkscan5*. The DNA binding motif for ZKSCAN5 is indicated above. The color scale represents the average expression level, and the size of the dot represents the percentage of cells expressing the gene. (**b**) *Left:* Coverage plot of the normalized snATAC-seq signal at the *Bnc2* locus. *Right:* Violin plot of *Bnc2* expression in E11.5-E13.5 XX and XY PGCs. (**c**) Joint Uniform Manifold Approximation

*Figure 2 continued on next page*

*Figure 2 continued*

and Projection (UMAP) showing *Bnc2* expression levels for E11.5-E13.5 XX and XY PGCs. Individual cells are color coded by the level of *Bnc2* expression. (**d**) Candidate factors (*x*-axis) regulating *Bnc2* expression based on their Cistrome Data Browser (Cistrome DB) regulatory potential score (*y*-axis). (**e**) Dotplot of the average expression and percentage of cells expressing the candidate factors that potentially regulate *Bnc2* expression. The color scale represents the average expression level, and the size of the dot represents the percentage of cells.

together, these findings provide insight into potential XY-specific regulatory mechanisms at the *Bnc2* locus in PGCs.

## Characterization of genes known for their roles in XX PGC development

We next performed the same analyses of XX PGC functional genes, *i.e.*, those that have been shown to be functionally important for XX PGC development (*Figure 3a*). Overall, as expected, we found that E13.5 XX PGCs expressed genes involved in XX germ cell development (i.e. *Rec8, Sycp2, Sycp3, Ctnnb1, Meioc, Rnf2, Stra8, Ythdc2,* and *Zglp1*) (*Spiller and Bowles, 2022*) substantially higher than E13.5 XY PGCs (*Figure 3a*). Moreover, *Rnf2* and *Zglp1* were strongly expressed at E12.5 in XX PGCs, and their expression occurred immediately prior to the activation of all other genes in the female cohort (*Figure 3a*). Although *Zglp1* was expressed in fewer XX PGCs than *Rnf2*, *Zglp1* expression increased from E12.5 to E13.5 whereas *Rnf2* expression decreases (*Figure 3a*). These data support previous studies that *Rnf2* and *Zglp1* control the timing of sexual differentiation of XX PGCs (*Nagaoka et al., 2020*; *Yokobayashi et al., 2013*). By comparison, the expression of *Ctnnb1* diminished in both XX and XY PGCs at E12.5, as expected given that negative regulation of WNT/β-catenin is required for XX PGCs to enter meiosis (*Le Rolle et al., 2021*; *Figure 3a*). In sum, these results provide insight into the timing, expression levels, and pervasiveness of female gene activation in XX PGC populations.

We next identified potential regulators of XX PGC functional genes using the same approach for XY PGCs (*Figure 2*). Our analysis identified significant enrichment of the binding motif for DMRT1 in the regulatory elements linked to XX PGC functional genes (*Figure 3a*). DMRT1 has been shown to control *Stra8* expression in a sex-specific manner, such that *Stra8* is activated in the fetal ovary and repressed in the fetal testis (*Krentz et al., 2011*). Our snRNA-seq data also demonstrated *Dmrt1* expression in both XX and XY PGCs (*Figure 3a*), which is consistent with previous findings (*Krentz et al., 2011*).

Since *Stra8* plays an essential role in controlling meiotic entry (*Spiller and Bowles, 2022*), we next sought to further understand the gene regulatory mechanisms underlying *Stra8* activation. First, we not only found that accessibility of the *Stra8* promoter was highest in XX PGCs, but also that *Stra8* chromatin accessibility was positively correlated with *Stra8* gene expression (*Figure 3b and c*). Second, the population of E13.5 XX PGCs displaying the strongest *Stra8* expression levels corresponded to the same population of XX PGCs with the highest module score of early meiotic prophase I genes (*Figure 3c*; *Appendix 1—figure 3a and b*). Next, to determine putative regulators of *Stra8* expression, we identified enriched TF binding motifs within 10 kb of the *Stra8* promoter using Cistrome DB (*Mei et al., 2017*; *Figure 3d*). Of the factors with the highest Cistrome DB Regulatory Potential Score, we observed XX-enriched expression of *Rxra, H2afy, Tcf12, Rad21, Smc3,* and *Smad3* (*Figure 3e*). Several of these factors, such as RAD21 and SMC3, are members of the cohesin complex, which serves both a structural and gene regulatory role during meiotic prophase I (*Mehta et al., 2013*; *Garcia-Cruz et al., 2010*). Moreover, RXRA has been shown to positively influence *Stra8* expression levels during oocyte development in the mouse (*Endo et al., 2019*). Taken together, these data provide a proof-of-principle example for the application of single-nucleus multiomics in identifying putative regulators of germ cell fate.

## Expression and in silico chromatin binding of RA receptors in PGCs

Since recent reports have demonstrated that meiosis occurs normally in the fetal ovary of mice lacking key members of the RA signaling pathway (*Vernet et al., 2020*), we next asked whether we could detect any evidence for gene regulation by RA receptors in PGCs. Previous studies have demonstrated that RA from the mesonephros induces meiosis in the fetal ovary whereas in the fetal testis, RA is degraded by the enzyme CYP26B1, therefore preventing PGC entry into meiosis (*Bowles*



**Figure 3.** Molecular characterization of genes known for their roles in female primordial germ cells (PGC) sex determination. (**a**) Dotplot of the average expression and percentage of cells expressing *Rec8, Sycp2, Sycp3, Ctnnb1, Meioc, Rnf2, Stra8, Ythdc2, Zglp1,* and *Dmrt1*. The DNA binding motif for DMRT1 is indicated. The color scale represents the average expression level, and the size of the dot represents the percentage of cells expressing the gene. (**b**) *Left:* Coverage plot of the normalized snATAC-seq signal at the *Stra8* locus. *Right:* Violin plot of *Stra8* expression in E11.5-E13.5 XX and XY PGCs. (**c**) Joint Uniform Manifold Approximation and Projection (UMAP) showing *Stra8* expression levels for E11.5-E13.5 XX and XY PGCs. Individual cells are color coded by the level of *Stra8* expression. (**d**) Candidate factors (*x*-axis) regulating *Stra8* expression based on their Cistrome Data Browser

*Figure 3 continued on next page*

Figure 3 continued
(Cistrome DB) regulatory potential score (*y*-axis). (**e**) Dotplot of the average expression and percentage of cells expressing the candidate factors that potentially regulate *Stra8* expression. The color scale represents the average expression level, and the size of the dot represents the percentage of cells.

*et al., 2018*; *Bowles et al., 2006*). However, this model was challenged by the findings that meiosis, and consequently oogenesis, still occurs in XX gonads of mice lacking the RA-synthesizing enzyme, ALDH1, or all RA receptors (*Vernet et al., 2020*; *Chassot et al., 2020*). Nevertheless, *Stra8* expression was significantly reduced in E13.5 fetal ovaries without *Aldh1a1-3* or all RA receptors (*Vernet et al., 2020*; *Chassot et al., 2020*), thus indicating that *Stra8* expression is likely controlled by the RA signaling pathway, in addition to others. We therefore investigated the expression and in silico chromatin binding patterns of members of the RA signaling pathway (*i.e.*, *Rara, Rarb, Rarg, Rxra, Rxrb, Rxrg, Crabp1,* and *Crabp2*) (*Napoli and Yoo, 2020*; *Appendix 1—figure 4a and b*). We found that expression of the RA receptor beta, *Rarb*, and cellular RA binding protein 1, *Crabp1*, were enriched in XX PGCs beginning at E12.5 and continuing to E13.5 (*Appendix 1—figure 4a and b*). Similarly, E13.5 XX PGCs showed enriched expression of the retinoid X receptor alpha, *Rxra*, and cellular RA binding protein 2, *Crabp2* (*Appendix 1—figure 4a and b*). By contrast, the expression of *Rara, Rarg,* and *Rxrb* was enriched in XY PGCs, and *Rxrg* was lowly expressed in both XX and XY PGCs (*Appendix 1—figure 4a and b*). We also observed significant enrichment of the 'motif activity score', an in silico measurement of the contribution of a specific motif to gene regulation (*Schep et al., 2017*), of all RA receptors in E13.5 XX PGCs (*Appendix 1—figure 4c and d*). These data indicate that the RA receptors display strong evidence of in silico chromatin binding in E13.5 XX PGCs but not E13.5 XY PGCs. Furthermore, the binding motifs for the RA receptors were predominantly located in E13.5 XX PGC chromatin peaks that were distal intergenic, intronic, or within promoters (*Appendix 1—figure 4e*; motifs for RARB and RXRA provided as examples). These data indicate that the binding motifs for the RA receptors are predominantly enriched in both distal and proximal regulatory elements in E13.5 XX PGCs. The RA receptor motifs were also statistically linked to genes involved in 'response to RA', 'negative regulation of cell cycle', and 'meiotic cell cycle' (*Appendix 1—figure 4f*; motifs for RARB and RXRA provided as examples). Finally, when we searched for the presence of RA receptor motifs in peaks linked to genes related to meiosis and female sex determination, we found that *Stra8, Rec8, Rnf2, Sycp1, Sycp2, Ccnb3,* and *Zglp1* contain the RA receptor motifs in their regulatory sequences (*Appendix 1—figure 4g*). Together, these in silico experiments indicate that the RA signaling pathway directly regulates the expression of several meiosis-related genes.

## Discovery of differentially expressed genes and differentially accessible regulatory loci underlying sex determination of PGCs

We further investigated our multiome dataset to discover genes and *cis*-regulatory elements enriched in PGCs in a sex-specific manner. First, we analyzed which annotated protein-coding genes showed transcriptional changes between XX and XY PGCs at each embryonic stage. Differential expression analysis ($\log_2$fold-change [FC]>0.25) of snRNA-seq data identified 87, 301, and 585 XX PGC-enriched genes and 75, 344, and 544 XY PGC-enriched genes at E11.5, E12.5, and E13.5, respectively (false discovery rate [FDR]-adjusted p-value<0.05; *Figure 4a*). The number of differentially expressed genes (DEGs) increased by ~3.5-fold during the transition from E11.5 to E12.5 PGCs and by 1.75-fold during the transition from E12.5 to E13.5 PGCs (*Figure 4a*). These findings demonstrate that the number of DEGs steadily increased over developmental time in both XX and XY PGCs.

An in-depth analysis of the DEGs between XX and XY germ cells identified sex-enriched genes for PGCs at each embryonic stage (*Appendix 1—figure 5a*). For example, we found elevated expression of the WNT protein regulator *Porcn* (*Madan et al., 2016*), the β-catenin binding protein *Grip1* (*Li et al., 2004*), and the growth regulator *Phlda2* (*Ma et al., 2020a*) in XX PGCs (*Appendix 1—figure 5a*). These observations are consistent with the fact that WNT/β-catenin signaling is involved in specifying the female fate of PGCs (*Spiller and Bowles, 2022*). We also identified XY-enriched DEGs, which included the histone demethylase *Kdm5d* (*Navarro-Costa et al., 2016*), the regulator of pluripotency *Dppa5* (*Klein and Knoepfler, 2021*), and the forkhead TF *Foxp1* (*Lam et al., 2013*; *Appendix 1—figure 5a*). Gene ontology (GO) analyses of DEGs revealed several biological processes that show sex-enriched expression patterns (*Appendix 1—figure 5b*). For example, the DEGs identified in XX PGCs were assigned to the 'meiotic chromosome organization', 'sister chromatid cohesion', and



**Figure 4.** Identification of differentially expressed genes (DEGs) and differentially accessible peaks in E11.5-E13.5 XX and XY primordial germ cells (PGCs). (**a**) Bar graph showing the number of DEGs between XX and XY PGCs at E11.5-E13.5. (**b**) Bar graph showing the number of differentially accessible chromatin peaks (DAPs) between XX and XY PGCs at E11.5-E13.5. (**c**) *Left:* Coverage plot of the normalized snATAC-seq signal at the DEG *Porcn* locus.

*Figure 4 continued on next page*

*Figure 4 continued*

Peak-to-gene linkages are indicated by the 'links' line. *Right:* Violin plot of *Porcn* expression in E11.5-E13.5 XX and XY PGCs. (**d**) *Left:* Coverage plot of the normalized snATAC-seq signal at the DEG *Rimbp1* locus. Peak-to-gene linkages are indicated by the 'links' line. *Right:* Violin plot of *Rimbp1* expression in E11.5-E13.5 XX and XY PGCs. (**e**) DAPI staining (gray) and immunofluorescence staining of TRA98 (pink), PORCN (yellow), and NR2F2 (blue) in E13.5 XX and XY gonads from C57Bl/6 and CD1 mice.

---

'WNT signaling pathway' gene ontologies (*Appendix 1—figure 5b*). The DEGs identified in XY PGCs were related to 'mitotic cell cycle', 'TGFβ signaling pathway', and 'regulation of translation' gene ontologies (*Appendix 1—figure 5b*). In sum, the DEGs between XX and XY PGCs were characterized by their sex-specific roles in meiotic/mitotic cell cycle regulation, post-transcriptional processing of mRNA, and cell-cell signaling properties (*Appendix 1—figure 5a and b*). These analyses provide a clear description of the transcriptional differences between XX and XY PGCs underlying the transition from bipotential germ cells to sex-specific oogonium or gonocytes.

To quantitatively assess chromatin dynamics during PGC sex determination, we next evaluated the number of differentially accessible chromatin peaks (DAPs) between XX and XY germ cells (*Figure 4b*). The DAP analyses ($log_2FC>0.25$) revealed 1720, 2311, and 5626 XX-enriched DAPs and 3341, 3117, and 15,546 XY-enriched DAPs at E11.5, E12.5, and E13.5, respectively (*Figure 4b*). The largest increase in the number of DAPs was found between XY PGCs (15,546) and XX PGCs (5,626) at E13.5. The DAPs for XX and XY PGCs at E11.5-E13.5 were predominantly located in promoter, distal intergenic, or intronic regions, indicating that these DAPs likely correspond to gene regulatory sites (*Appendix 1—figure 5c*). Together, these data suggest that the chromatin architecture of PGCs, and especially the chromatin of XY PGCs, undergoes significant remodeling during sex determination.

We next sought to associate sex-specific changes in chromatin accessibility with changes in gene expression for PGCs. To determine peak-gene associations, we correlated snATAC-seq peak accessibility with the expression of nearby genes, with the goal to identify sex- and stage-specific chromatin peaks positively correlated with mRNA expression (p-value<0.05; z-score>0). This analysis detected putative gene regulatory interactions based on correlations between chromatin peak accessibility and gene expression levels for all peaks and genes within 500 kb of each other (*Ma et al., 2020b*). Given the notable increase in DAPs between XX and XY PGCs at E13.5 (*Figure 4b*), we first identified the predicted target genes of all sex-specific DAPs at this stage (*Appendix 1—figure 6a*). For E13.5 XX PGCs, we observed positive DAP-gene correlations for genes enriched for biological functions such as 'WNT signaling pathway', 'chromosome segregation', 'chromatin organization', 'meiotic cell cycle', and 'RNA stabilization' (*Appendix 1—figure 6a*). A similar analysis of E13.5 XY PGCs revealed positive DAP-gene correlations for genes involved in 'ncRNA metabolic process', 'mitotic cell cycle', 'DNA repair', 'regulation of translation', and 'RNA localization' (*Appendix 1—figure 6b*). These findings demonstrate that we can use chromatin accessibility information to identify the regulatory elements linked to sex-specific genes in PGCs.

We next investigated the associations of differentially accessible (DA) chromatin peaks with DEG expression for each PGC type. At E11.5, we found that DA peak-to-gene linkages were associated with ~20% of DEGs in XX PGCs and ~1% of DEGs in XY PGCs (*Appendix 1—figure 6c*). The overlap of DA peak-to-gene linkages with DEGs increased steadily at E12.5 and E13.5, with ~35% and~60% of DA peaks in E13.5 XX and XY PGCs, respectively, positively correlated with DEG expression (*Appendix 1—figure 6c*). For example, at E13.5, we identified the DA peaks positively correlated with the DEGs *Porcn* (involved in the processing of WNT proteins) (*Madan et al., 2016*) and *Rec8* (involved in meiotic recombination) (*Spiller and Bowles, 2022*) in XX PGCs and *Rimbp1* (regulates cytosolic calcium ion concentration) (*Mencacci et al., 2021*) in XY PGCs (*Figure 4c and d*; *Appendix 1—figure 6d*). Immunofluorescence staining also confirmed strong expression of *Porcn* in E13.5 fetal XX germ cells and diminishing expression of *Porcn* in E13.5 fetal XY germ cells (*Figure 4e*), demonstrating that single-nucleus multiomics can identify DEGs and the DA chromatin peaks that are associated with their expression.

# A computational approach to identify candidate TFs regulating XX PGC gene expression

To better utilize the multiomics data, we developed an analytical flow to identify potential TFs that are involved in regulating the expression of DEGs in PGCs. First, we classified peak status by identifying peaks that are differentially accessible between XX and XY PGCs following the analytical framework in *Figure 5a(i)*. Second, we linked peaks to genes to identify positive correlations between DA chromatin peaks and the expression of DEGs at each embryonic stage (p-value<0.05; z-score>0; *Figure 5a(ii)*). Third, we identified all enriched TF binding motifs located within DA chromatin peaks and found unique motifs with a significant FDR threshold (p-value<0.05; *Figure 5a(iii)*). We investigated only the TFs that were differentially expressed between sexes and that showed germline enrichment. Finally, we predicted the target genes of candidate TFs by determining the genomic position of their enriched motifs and the genes linked to their resident peaks (*Figure 5a(iv)*). This analytical flow allowed us to rank enriched TF binding motifs based on variation in chromatin accessibility and unbiasedly infer TFs involved in PGC sex determination with greater confidence.

In XX PGCs, we uncovered several candidate TF regulators of E11.5-E13.5 XX-enriched DEGs (*Appendix 1—figure 7a–c*). We plotted the mRNA expression levels of TFs and the *in silico* chromatin binding score, termed 'motif activity', of all significantly enriched binding motifs (*Appendix 1—figure 7a–c*). For example, in E11.5 XX PGCs, RREB1 is a differentially expressed TF showing high gene expression, but negative levels of in silico chromatin binding (*Appendix 1—figure 7a*). This finding indicates that although RREB1 is a highly expressed TF in E11.5 XX PGCs, RREB1 does not likely promote increased chromatin accessibility either because it does not bind to chromatin or a critical cofactor is unavailable (*Appendix 1—figure 7a*). Following this analytical framework, we focused our analysis on TFs that showed high levels of gene expression, were differentially expressed in XX PGCs, and displayed a positive in silico chromatin binding score (*Appendix 1—figure 7a–c*). The TFs meeting these criteria were KLF12, NR6A1, and ZFX in E11.5 XX PGCs; TFAP2c in E12.5 XX PGCs; and BACH2, CREM, GATA2, GLI3, MGA, NFKB1, PLAGL1, TBX4, TCFL5, TFAP2c, ZBTB7c, and ZFX in E13.5 XX PGCs (*Appendix 1—figure 7a–c*). Of these 16 TF candidates with PGC-enriched transcript expression, only 7 (*Gata2, Mga, Nr6a1, Tbx4, Tcfl5, Tfap2c,* and *Zfx*) showed enriched expression in germ cells when compared to the somatic compartment of the gonad and, thus, met our requirements for further *in silico* investigation (*Appendix 1—figure 8a*). These candidate TFs have reported roles in cell fate determination, growth and fertility regulation, and meiotic progression (*Galán-Martínez et al., 2022*; *Kojima et al., 2021*), all of which are in line with the cellular events that occur as PGCs commit to the female pathway. Nonetheless, the chromatin binding and gene expression of GATA2, MGA, TBX4, TCFL5, and TFAP2c showed the clearest patterns of sexual dimorphism in E13.5 PGCs and represented the most compelling TF candidates (*Figure 5b and c*; *Appendix 1—figure 8b–f*).

The binding motif for TFAP2c (also known as *AP2γ*) showed prominent TF-associated accessibility, especially when compared to all other significantly enriched TF binding motifs (*Appendix 1—figure 7b and c*). Immunofluorescence staining of TFAP2c also showed enrichment of TFAP2c in E13.5 XX PGCs when compared to the somatic compartment and E13.5 XY PGCs (*Appendix 1—figure 9a*). Compared to XY PGCs, XX PGCs had enriched *Tfap2c* expression (p-value<0.05; $\log_2$FC >0.25) and enriched accessibility of the TFAP2c motif at E12.5-E13.5 (*Figure 5b*). The TFAP2c motif was close to genes related to glycogen metabolism, carbohydrate metabolism, and chromatin modification (*Figure 5d*). The predicted target genes of TFAP2c accounted for 75% (372/494) of all DEGs with peak-to-gene linkages in XX PGCs (*Figure 5e*), suggesting that TFAP2c is likely a high-level regulator of female PGC sex determination.

In addition to TFAP2c, we also investigated the regulatory potential of TCFL5 (Transcription Factor Like 5) in greater detail. TCFL5 was the most differentially expressed in E13.5 XX PGCs (p-value<0.05; $\log_2$FC>0.25) (*Figure 5c*). Accessibility of the TCFL5 motif was also enriched in E13.5 XX PGCs, indicating that chromatin binding of TCFL5 is likely strongest at this stage (*Figure 5c*). The predicted target genes of TCFL5 totaled 74% (367/494) of all DEGs with peak-to-gene linkages in XX PGCs (*Figure 5e*). A large majority of the predicted target genes of TCFL5 were also predicted to be the target genes of the enriched TFs presented in *Figure 5e*, *e.g.*, the predicted target genes of these TFs overlapped with 4–100% of the predicted target genes of TCFL5. The TCFL5-associated genes were related to chromosome segregation, meiotic cell cycle, sister chromatid cohesion, histone modification, regulation of TF activity, and cell-cell signaling by WNT. Our findings are supported by a recent

**Figure 5.** Analysis of gene networks in E11.5-E13.5 XX and XY primordial germ cells (PGCs). (**a**) Illustration of the analytical flow to infer gene networks (i.e. transcription factors [TFs] and their predicted target genes) involved in PGC sex determination. (**b–c**; **i–j**) Line plots showing the expression levels and in silico chromatin binding, termed 'motif activity', of *Tfap2c* (**b**), *Tcfl5* (**c**), *Foxk2* (**i**), and *Pou6f2* (**j**) in E11.5-E13.5 XX (pink) and XY (teal) PGCs. The p-value of motif enrichment and the TF binding motif are indicated. (**d and k**) Enrichment analysis of biological process gene ontology (GO) of the predicted target genes for TFAP2c (**d**) and FOXK2 (**k**). The color scale represents the p-value of GO term enrichment, and the size of the dot indicates the gene ratio for each GO term. (**e and l**) The number of predicted target genes for enriched TFs in XX PGCs (**e**) and XY PGCs (**l**). (**f**) Circos plot

*Figure 5 continued on next page*

*Figure 5 continued*

showing whether the motifs for TFAP2C, TCFL5, ZFX, MGA, or NR6A1 (indicated in black) are detected in the regulatory loci linked to *Tbx4, Nr6a1, Mga, Tcfl5,* or *Tfap2c* (indicated in gray). (**m**) Venn diagram showing the overlap of predicted target genes for POU6F1/2, FOXK2, and FOXO1. (**g–h**) Icon array showing the presence (pink/purple) or absence (light gray) of the motifs for TFAP2c, TCFL5, ZFX, MGA, NR6A1, TBX4, and GATA2 in peaks linked to genes related to meiosis (**g**) or WNT signaling (**h**). (**n–o**) Icon array showing the presence (teal) or absence (light gray) of the motifs for FOXO1, FOXK2, and POU6F1/2 in peaks linked to genes related to mitosis (**n**) or signal transduction (**o**).

report on the role of TCFL5 in XX PGCs, which demonstrated the requirement of TCFL5 in meiotic progression and female fertility (*Zhang et al., 2023*).

Given that the motifs for TFAP2c and TCFL5 were not detected at every XX-enriched DEG, we hypothesize that a network of TFs, such as those identified in *Appendix 1—figure 8a*, could work in cooperativity to regulate XX PGC gene expression. For example, we found that the predicted target genes of ZFX, MGA, NR6A1, TBX4, and GATA2 accounted for 73% (359/454), 31% (152/494), 20% (100/494), and 6% (32/494), respectively, of all DEGs with peak-to-gene linkage in XX PGCs (*Figure 5e*). We also found that TFAP2c, TCFL5, ZFX, MGA, and NR6A1 may regulate the expression of each other (*Figure 5f*). For instance, the motif for ZFX was located in the enhancers or promoters for TFAP2c, TCFL5, MGA, and NR6A1 (*Figure 5f*). Likewise, NR6A1 is predicted to control the expression of TFAP2C, TCFL5, and TBX4 (*Figure 5f*). Furthermore, the motifs for the candidate TFs were statistically linked to genes involved in 'meiotic cell cycle' and the 'WNT signaling pathway' (*Figure 5g and h*). Meiosis-related genes such as *Tex15, Sycp1, Dmc1,* and *Hormad1* contained the motifs for at least three of the predicted TFs in their regulatory sequences (*Figure 5g*). A similar pattern was also found for genes related to WNT signaling, e.g., *Pkd1, Notum, Porcn, Fzd3,* and *Eda* were predicted to be regulated by the binding of at least two of the putative TFs in XX PGCs (*Figure 5h*). These data indicate that there may be unknown and undefined TF regulatory networks underlying XX PGC identity and function.

## Identification of candidate TF regulators of XY PGC differentiation

To identify candidate TF regulators involved in XY PGC sex determination, we performed the same analysis as described for XX PGCs (*Figure 5a*). Our analysis focused strictly on differentially expressed TFs that show evidence of chromatin binding based on the empirical *in silico* chromatin binding score (*Appendix 1—figure 10a–c*). In contrast to XX PGCs (*Appendix 1—figure 7*), we did not identify any significantly enriched TF binding motifs (p-value>0.05) meeting our criteria in E11.5 XY PGCs (*Appendix 1—figure 10a*). Only three XY-enriched TFs, FOXP1, POU6F2, and TCF4, were differentially expressed between XY and XX PGCs at E12.5 (*Appendix 1—figure 10b*), and only TCF4 showed a positive in silico chromatin binding score in E12.5 XY PGCs (*Appendix 1—figure 10b*). For E13.5 XY PGCs, we detected five significantly enriched motifs that bound differentially expressed TFs (p-value<0.05): FOXK2, FOXO1, FOXP1, POU6F1, and TCF4 (*Appendix 1—figure 8c*). Of these six XY-enriched TF candidates from E11.5 to E13.5, FOXO1, FOXK2, and POU6F1/2 emerged as the top candidates based on germline-enriched expression when compared to somatic cells, differential expression in XY PGCs, and positive *in silico* chromatin binding scores (*Appendix 1—figure 11a and b*; *Figure 5i and j*). Together, these data indicate that XY PGCs rely less on differentially expressed TF-mediated gene regulation when compared to XX PGCs.

At E13.5, when the most TFs and their motifs were found in XY PGCs, we observed significant enrichment of the FOX (forkhead-box) protein family of TFs (*Appendix 1—figure 10c*). FOX proteins contain pioneering transcripti properties that allow for binding to condensed chromatin and enabling competency of gene activity through histone remodeling (*Zaret, 2020*). Given their role in opening condensed chromatin, FOX TFs play important roles in regulating the expression of genes involved in cellular differentiation, proliferation, plasticity, and growth (*Herman et al., 2021*). In particular, we found that the expression of *Foxo1* and *Foxk2* were specific to XY PGCs when compared to XX germ cells and the XY somatic compartment (*Appendix 1—figure 10c*; *Appendix 1—figure 11a and b*; *Figure 5i*). From our dataset, we found the FOXK2 motif positively associated with genes related to cellular proliferation, cell cycle arrest, mitosis, and DNA damage checkpoint (*Figure 5k*). In addition, the predicted target genes of FOXK2 totaled 26% (54/208) of all DEGs with peak-to-gene linkages in XY PGCs (*Figure 5l*). This finding was consistent with the number of predicted target genes of FOXO1, which included 28% (58/208) of all DEGs with peak-to-gene linkages in XY PGCs and may be

the result of motif redundancy among FOX proteins. Given the substantial increase in the number of DA peaks in E13.5 XY PGCs (*Figure 4b*), we hypothesize that the pioneering properties of the FOX family promotes chromatin remodeling as XY PGCs transition to gonocytes.

We also detected significant enrichment of the POU domain, class 6 (POU6) TF family in XY PGCs (*Appendix 1—figure 10b and c*; *Appendix 1—figure 11a and c*; *Figure 5j*). The POU family plays important roles in regulating cellular differentiation and the timing of cellular events (*Xiao et al., 2022*). We found XY-enriched expression of POU6F1 and POU6F2 in PGCs, however, only POU6F2 displayed germline enrichment when compared to the somatic cell populations of the gonad (*Appendix 1—figure 11a and c*; *Figure 5j*). Furthermore, we observed the strongest POU6F1 and POU6F2 in silico chromatin binding scores in XY PGCs at E13.5 (*Appendix 1—figure 11c*; *Figure 5j*). The POU6F1/2 motif was associated with 19% (40/208) of all DEGs with peak-to-gene linkages in XY PGCs (*Figure 5l*). POU6F1/2-associated genes were related to voltage-gated chloride channel activity (*Ano6*) (*Yu et al., 2015b*), regulation of cell proliferation (*Appl1*) (*Zhao et al., 2010*), regulation of actin filaments (*Fmnl2*) (*Pan et al., 2020*), Notch signaling (*Maml3*) (*Oyama et al., 2011*), lipid metabolism (*Osbpl9*) (*Lehto and Olkkonen, 2003*), and ubiquitin ligase activity (*Usp7*) (*Turnbull et al., 2017*; *Figure 5m*). Thus, the POU6 TF family is a top candidate to coordinate gene expression activity in XY PGCs.

We next compared the number and identity of FOXO1-, FOXK2-, and POU6F1/2-associated genes in XY PGCs. First, we did not detect the FOXO1, FOXK2, or POU6F1/2 motifs in the promoters/enhancers for each of these TFs. These findings indicate that these TFs do not regulate the expression of each other, and other mechanisms or regulatory pathways are required to activate the expression of candidate TFs in XY PGCs. Furthermore, the predicted target genes of the FOX and POU6 TF families shared little overlap with each other (11/87) (*Figure 5m*). The genes that were shared between FOXO1, FOXK2, and POU6F1/2 had reported functions in maintaining survival after DNA damage (*Alkbh8*) (*Zheng et al., 2014*), TGFβ signaling (*Bmp7*) (*Jia and Meng, 2021*), and initiation of mitosis (*Nek7*) (*Bachus et al., 2022*; *Figure 5m*). By comparison, the predicted target genes of FOXO1 and FOXK2 showed considerable overlap (42/47) and included genes related to RNA splicing (*Aqr*) (*Li, 2020*), cell cycle progression (*Cdk6*) (*Tigan et al., 2016*), cellular metabolism (*Eno1* and *Odc1*) (*Song et al., 2023*; *Choi et al., 2016*), cell death (*Pml*) (*Quignon et al., 1998*), and negative regulation of the IGF1 signaling pathway (*Socs2*) (*Farquharson and Ahmed, 2013*; *Figure 5m*). The motifs for FOXO1, FOXK2, and POU6F1/2 were statistically associated with genes involved in 'mitotic cell cycle' and 'signal transduction' (*Figure 5n and o*). Mitosis-related genes such as *Cradd, Cdk6, Exoc6b,* and *Pml* were potentially regulated by FOXO1 and FOXK2 (*Figure 5n*). In addition, we noted that the POU6F1/2 motif was associated with *Cradd* and *Psmg2* (*Figure 5n*). We observed a similar pattern for genes related to signal transduction (*Figure 5o*). For example, *Bmp7, Map3k5, Akt3,* and *Sulf1* were associated with FOXO1 and FOXK2, whereas *Bmp7, Map3k5,* and *Sfprp1* were associated with POU6F1/2 motif activity (*Figure 5o*). Taken together, these data suggest that in XY PGCs, although FOXO1, FOXK2, and POU6F1/2 regulate genes with similar properties, the FOX and POU6 TF families are unlikely to collaboratively regulate gene expression.

## Sex-specific enrichment of ligand-receptor signals between gonadal support cells and PGCs

Sexual dimorphism of germ cell differentiation is thought to be controlled by a meiosis-inducing substance produced by the somatic cells of the fetal ovary and/or by a meiosis-preventing substance produced by the fetal testis (*McLaren and Southee, 1997*). In experimentally generated XX/XY chimeras, both XX and XY PGCs can initiate meiosis when residing in the fetal ovary or enter mitotic arrest when colonizing the fetal testis (*Evans et al., 1977*). In the fetal ovary, PGC meiosis is induced by RA produced by ovarian and mesonephric somatic cells (*Spiller and Bowles, 2022*). Consistent with this model, RA signaling induces expression of *Stra8* and *Meiosin*, both of which are critical for promoting entry into meiosis (*Spiller and Bowles, 2022*). Signaling by WNT and BMP also influences the timing of meiotic onset in XX PGCs (*Spiller and Bowles, 2022*), and inhibin/activin signaling regulates XX germ cell growth and proliferation (*da Silva et al., 2004*). In the fetal testis, RA degradation by CYP26B1 and FGF9 signaling from XY supporting cells result in reduced *Stra8* expression, consequently blocking meiotic entry in XY PGCs (*Spiller and Bowles, 2022*). Thus, PGC sex determination is dependent on external signals from the gonadal environment rather than chromosomal sex.



**Figure 6.** Sexually dimorphic interactions between primordial germ cells (PGCs) and gonadal supporting cells. (**a**) Illustration of cell communication between PGCs and gonadal supporting cells. (**b**) Number of significant interactions or ligand-receptor pairs (p-value<0.05) between supporting cells and PGCs in E11.5-E13.5 XX and XY gonads. (**c**) Number of significantly enriched ligand-receptor pairs, or interactions, per signaling pathways induced by either WNTs, inhibins/activins, or BMPs in E13.5 XX and XY gonads. (d, f, and h) Venn diagrams showing the overlap of significant interactions between supporting cells and PGCs in XX and XY gonads at E11.5 (**d**), E12.5 (**f**), and E13.5 (**h**). (e, g, and i) Dotplots showing the mean expression for all interacting partners in representative ligand-receptor pairs in XX and XY gonads at E11.5 (**e**), E12.5 (**g**), and E13.5 (**i**). Ligand-receptor pairs in pink represent XX-specific interactions, ligand-receptor pairs in blue represent XY-specific interactions, and ligand-receptor pairs in gray represent interactions found in both XX and XY gonads. The size of the dot represents the mean expression value.

To uncover potential instructive signals from the supporting somatic cells (Sertoli cells in the fetal testis or granulosa cells in the fetal ovary) to PGCs, we searched for supporting cell-derived ligands and the corresponding receptors in PGCs (*Figure 6a*), using CellphoneDB v5.0 for cell communication analysis (*Troulé, 2023*). We first identified the supporting cell clusters from the snRNA-seq dataset of whole fetal gonads using the established female and male supporting cell markers *Runx1* and *Sox9*, respectively (*Stévant et al., 2019*; *Nicol et al., 2019*). We next quantified the number of significant interactions between supporting cells and PGCs at each embryonic stage (p-value<0.05; *Figure 6b*).

Such analyses revealed that the number of interactions gradually increased from E11.5 to E13.5 for both XX and XY gonads (*Figure 6b*). We also observed that the number of ligand-receptor pairs was 15–38% higher in XX gonads, compared to XY gonads at the same embryonic stage (*Figure 6b*). To validate our computational approach, we next searched for known pathways associated with PGC differentiation, such as signaling by WNT, inhibin/activin, and BMP (*Figure 6c*). The number of significant interactions per pathway increased in both XX and XY gonads from E11.5 to E13.5 (*Figure 6c*; *Appendix 1—figure 12a and b*). We also observed a moderate increase in the number of interactions for the inhibin/activin and BMP pathways in XX gonads when compared to XY gonads at the same embryonic stage (*Figure 6c*). Finally, we found that the number of interactions related to WNT signaling were nearly two to three times greater in XX than XY gonads (*Figure 6c*; *Appendix 1—figure 12a and b*). Together, these data demonstrate that sexually dimorphic patterns of cell signaling events were present between supporting cells and PGCs.

We next searched for new sex-specific enrichment of ligand-receptor pairs between supporting cells and PGCs (*Figure 6d–i*). For each embryonic stage, we first identified which signaling pathways were detected in XX alone, XY alone, or both XX and XY gonads that met our statistical threshold (p-value<0.05; *Figure 6d, f, and h*). At E11.5, XX and XY gonads shared the greatest overlap of supporting cell-derived ligands and PGC-expressing receptors (*Figure 6d*). The shared pathways between XX and XY gonads were largely related to adhesion by collagen/integrin, fibronectin, and cadherin. E11.5 XX gonads had enrichment of 19 ligand-receptor pairs related to, for example, BMP, ephrin, integrin, FGF, and WNT signaling (*Figure 6d*). By contrast, E11.5 XY gonads had significant enrichment of 10 ligand-receptor pairs predominantly related to cell adhesion pathways like ephrin, nectin, and integrin (*Figure 6d*). To highlight the general trends of expression between sexes, we compared the mean expression of all representative ligand-receptor pairs from each group in *Figure 6d, f, and h*; *Figure 6e, g, and i*. In E11.5 gonads, we observed dimorphic expression of XX-enriched interactors, such as BMP2/BMR1A/AVR2B and WNT4/FZD10/LRP5, but not the XY-enriched interactors CADM1/NECTIN3 and EFNB1/EPHB3 (*Figure 6e*). Furthermore, we did not observe a sex bias for the mean expression of interacting pairs for the shared pathway DLK1:NOTCH3 between E11.5 XX and XY gonads (*Figure 6e*). These data indicate that at E11.5, sexual dimorphism of signaling pathways, in terms of number and mean expression, is most prominent between XX supporting cells and XX PGCs.

In E12.5 and E13.5 gonads, we observed a stronger XX bias in ligand-receptor interactions between supporting cells and PGCs (*Figure 6f–i*). At E12.5, we observed 49 and 24 significant interactions in XX and XY gonads, respectively (*Figure 6f*). In E12.5 XX gonads, BMP, insulin-like growth factor (IGF), R-spondin, and WNT signaling were the most frequently observed pathways and displayed strong XX-enriched mean expression (*Figure 6f and g*). In XY gonads, detection of new signaling pathways emerged from E11.5 to E12.5, including signaling by BMP6, chemokines, Matrix metalloproteinases (MMP), and Neuregulin (*Figure 6f*). Compared to E12.5, E13.5 XX gonads continued to have enrichment for the BMP, IGF, and WNT signaling pathways with additional pathways related to midkine (MDK), Calsyntenin (CLSTN), and SLIT and NTRK-like protein signaling detected (*Figure 6f and h*). The BMP5:BMR1A:AVR2B and MDK:PTPRZ1 ligand-receptor interactors also showed XX-biased mean expression (*Figure 6i*). Similarly in E12.5 and E13.5 XY gonads, we observed maintenance of the ligand-receptor pairs, BMP6:ACVR:1A2B and BMP6:BMR1A:AVR2B (*Figure 6f and h*). However, enrichment of signaling by Desert hedgehog, Notch, and WNT inhibition became significant in E13.5 XY gonads but not E12.5 (*Figure 6f and h*). High mean expression of DHH:PTCH:GAS1 and DLK1:NOTCH3 interactors was detected in E13.5 XY gonads but became indetectable in E13.5 XX gonads (*Figure 6i*). Taken together, these data suggest that sex-specific enrichment of ligand-receptor interactions become gradually more apparent from E11.5 to E13.5.

## Discussion

Male and female germ cells carry the hereditary information required for the generation of a new individual and the perpetuation of sexually reproducing species. Errors in germ cell development and differentiation can result in a variety of reproductive diseases, such as infertility (*Czukiewska and Chuva de Sousa Lopes, 2022*). Disruption of gametogenesis during early embryonic development can even lead to the onset of disease later in life, a concept which is supported by the Developmental Origins of Health and Disease (DOHaD) theory (*Jazwiec and Sloboda, 2019*). Thus, it is of critical

importance to understand the molecular and epigenetic signatures of developing germ cells and particularly PGCs. Nonetheless, the heterogeneous nature and low cell numbers of PGC populations often hinder the identification of novel factors that influence PGC fate. To overcome this limitation, we generated the first single-nucleus atlas of gene expression and chromatin accessibility for murine PGCs during sex determination. We demonstrate that single-nucleus multiomics successfully captures PGC sex determination at high resolution, identifies putative TFs involved in PGC sex determination, and provides insight into the cellular communication pathways between PGCs and the supporting cells of the gonads.

## Integrated RNA and ATAC data for individual PGCs identified significant heterogeneity in XX PGC populations compared to XY PGCs

We first identified the subgroupings of E11.5-E13.5 XX and XY PGCs and resolved the patterns of gene expression and chromatin accessibility underlying each PGC population. We observed that the combined analysis of integrated snRNA- and snATAC-seq data produced better clustering and separation of PGCs than either assay alone. For example, we found that at E11.5, XX and XY PGCs share highly similar transcriptomic and open chromatin profiles, indicating that PGCs at this stage are bipotential at the level of gene regulation and chromatin accessibility. This observation agrees with the study that showed the global overlap of transcriptomes between E11.5 PGCs from both sexes, in which they detected only 101 DEGs between XX and XY PGCs with functional annotations related to chaperonin complexes and proteasome formation (*Mayère et al., 2021*). At E12.5, subpopulations of PGCs clustered separately within each sex, supporting the model that PGCs do not commit to a sex-specific fate in a synchronous manner (*Soygur et al., 2021*). At E13.5, two subclusters of XX PGCs appear, consistent with the asynchronous entry of XX PGCs into meiosis (*Soygur et al., 2021*). Others also reported substantial heterogeneity of human and mouse PGC populations, often arising from intrinsic variation in the epigenetic reprogramming required for sex differentiation (*Chen et al., 2022*; *Nguyen et al., 2019*). These data underscore the discovery potential of simultaneously measuring multiple layers of gene regulation to define the developmental fate of PGCs.

## Molecular signatures of differentiating PGCs revealed disease-related genes associated with PGC clusters

When examining genes that could be responsible for the expression-based clustering of PGCs, we found TFs, cell cycle regulators, chromatin modifiers, and ribosome biogenesis genes underlying the diverse transcriptional profiles of differentiating PGCs. Several of the expressed genes enriched in individual PGC clusters were involved in infertility or disease processes when nonfunctional. For example, mutations in *Lars2*, the mitochondrial leucyl-tRNA synthetase, is associated with premature ovarian failure in women (*Pierce et al., 2013*) and aberrant methylation at the promoter of *Mgmt*, a gene that is required for DNA repair and linked to the formation of testicular germ cell tumors in humans (*da S Martinelli et al., 2017*). The dynamic and varied regulatory profiles of PGCs that we observed highlight the necessity of pairing chromatin accessibility information with gene expression data for elucidating the mechanisms underlying PGC sex determination.

## Sexual dimorphism in chromatin accessibility increases temporally during PGC development

The chromatin landscape of PGCs is the most sexually dimorphic at E13.5 when compared to E11.5 and E12.5. The substantial increase in DAPs at E13.5 leads to several hypotheses that could account for the changes to PGC chromatin architecture during sex determination. First, it is possible that the chromatin of bipotential E11.5 PGCs is primed for activating either sex-specific differentiation pathway. Chromatin priming occurs when chromatin is maintained in an accessible state prior to gene expression and is evidenced in a number of developmental contexts, such as during specification of hematopoietic cells and lineage commitment of embryonic stem cells (*Bonifer and Cockerill, 2017*; *Tee and Reinberg, 2014*). During development, chromatin priming fine-tunes gene activation by enabling a rapid response to external cues that induce cell differentiation (*Bonifer and Cockerill, 2017*). Second, we posit that E11.5 PGCs have fewer DA peaks because they are actively dividing. Mitotic chromosome condensation is known to induce a state of heterochromatin and the eviction of TFs from chromatin in actively dividing cells (*Raccaud and Suter, 2018*). Third, the increased number

of DA peaks at E13.5 may be the result of changes to chromatin structure as XX PGCs enter meiotic prophase I, an event that requires dramatic changes to meiotic chromosomes as they undergo DNA double-strand breaks, synapsis, and crossing over (*Gray and Cohen, 2016*). Indeed, increased chromatin accessibility was found in mouse spermatocytes during prophase I of meiosis (*Wiltshire et al., 1998*). Finally, the upregulated expression of forkhead TFs in male PGCs at E12.5 and E13.5 may induce the *de novo* establishment of DA peaks through their pioneering activity. Nonetheless, our results are consistent with previous reports that demonstrate the global decrease in DNA methylation levels or increased chromatin accessibility, in mouse PGCs beginning at E9.5 and continuing through E13.5 (*Seisenberger et al., 2012*).

## Single-nucleus multiomics identifies evidence for chromatin priming during PGC lineage commitment

Moreover, not all sex- and stage-specific DA chromatin peaks account for changes in differential gene expression for PGCs. Chromatin accessibility and gene expression are often asynchronously correlated during cellular differentiation, such as during chromatin priming (*Ma et al., 2020b*). For example, accessibility of regulatory loci involved in cell fate decisions may precede gene expression to prime cells toward one developmental lineage vs another (*Ma et al., 2020b*). In the context of germ cell differentiation, PGCs must also undergo extensive chromatin remodeling events, including genome-wide DNA demethylation and a reduction in both repressive and activating histone modifications, prior to sex determination (*Huang et al., 2021*). Indeed, chromatin priming has been observed in PGCs from other species, such as macaques, pigs, and goats (*Chen et al., 2022*). Thus, it is likely that the DA chromatin peaks observed for E11.5 and E12.5 PGCs in mice may either reflect global chromatin remodeling events associated with sex determination or foreshadow future cell fate decisions by remaining poised for TF binding. Nevertheless, a comprehensive profile of 3D chromatin structure and enhancer-promoter contacts in differentiating PGCs is needed to fully understand how changes to chromatin facilitate PGC sex determination.

## The gene regulatory networks of XX PGCs are enriched for the TFs, TFAP2c, TCFL5, GATA2, MGA, NR6A1, TBX4, and ZFX

Our multiomics approach uncovered compelling TF candidates that may be involved in PGC development and differentiation. In XX PGCs, seven TF prospects were identified: GATA2, MGA, NR6A1, TBX4, ZFX, TCFL5, and TFAP2c. Of these TFs, TCFL5 and TFAP2C showed the strongest enriched expression in XX PGCs. TCFL5 binds to the promoters of meiotic genes such as *Sycp1* (synapsis) and *Stag3* (cohesion) during spermatogenesis in the mouse (*Galán-Martínez et al., 2022*). *Tcfl5* is indispensable for the completion of meiosis and spermatid maturation (*Galán-Martínez et al., 2022*) in male germ cells and meiosis of female germ cells (*Zhang et al., 2023*). Our multiomics data also demonstrate strong *Tfap2c* expression and motif enrichment in XX PGCs beginning at E12.5 and continuing through E13.5. These findings are corroborated by other single-cell RNA-seq analyses that showed *Tfap2c* expression in human and mouse PGCs and gonocytes (*Kojima et al., 2021*; *Laronda, 2023*; *Chen et al., 2019*; *Schemmer et al., 2013*; *Schäfer et al., 2011*). Additionally, TFAP2c has several reported roles in cell-type specification in both mouse and human, including germline specification and maintenance of germ cell identity (*Kojima et al., 2021*), remodeling of naïve enhancers (*Pastor et al., 2018*), and hormone responsiveness (*Woodfield et al., 2007*). Loss of *Tfap2c* in mice also leads to a reduction in germ cell number by either PGC apoptosis or PGC differentiation into somatic cells (*Weber et al., 2010*). Considering that changes in gene regulation leading to the female and male pathways begin as early as E12.5 (*Spiller and Bowles, 2015*), understanding how TCFL5 and TFAP2c activity is induced and investigating the role of TFAP2c in PGC sex determination are important future directions.

## XY PGCs showed sex-specific enrichment of TFs associated with reproductive dysfunction

In XY PGCs, we detected significant enrichment of the forkhead-box and POU6 families of TFs. In particular, FOXO1, FOXK2, and POU6F1/2 exhibited XY biased expression and in silico chromatin binding. Although the function of *Foxo1* has been studied in gonocytes (*Goertz et al., 2011*), little is known about the roles of *Foxk2* and *Pou6f1/2* in PGCs. Loss of *Foxo1* in embryonic gonocytes

leads to defective spermatogenesis in adulthood (*Goertz et al., 2011*), and inactivation of *Foxo1* in spermatogonia results in germ cell loss and male infertility (*Shen et al., 2022*). FOXO1 activity is also widely involved in regulating metabolic homeostasis, redox balance, cell death, and DNA damage repair (*Xing et al., 2018*). Likewise, FOXK2 functions in similar roles as FOXO1, such as in cellular metabolism, cell survival, and apoptosis (*Kang et al., 2022*). Furthermore, the POU6 family consists of POU6F1 and POU6F2, which share highly conserved DNA motif recognition sequences and function in similar cellular pathways (*Malik et al., 2018*). Both POU6F1 and POU6F2 have been reported to suppress tumor proliferation (*Yoshioka et al., 2009*), and POU6F2 mutations have been implicated in the progression of Wilms tumor, hypogonadism, and pubertal failure in human patients (*Cho et al., 2023*). Investigating the cooperativity of these TFs will greatly improve our understanding of how mitotic arrest and the commitment to spermatogenesis are regulated during germ cell development.

## Single-nucleus RNA-seq maps the temporal expression patterns of WNT, BMP, and RA signaling during PGC sex determination

During sex determination, PGCs respond to signals from the supporting cells of the gonads to determine their sex-specific fate. In the fetal ovary, *Runx1+* supporting cells produce WNT4 to maintain germ cell pluripotency and prevent precocious meiotic entry (*Spiller and Bowles, 2022*; *Le Rolle et al., 2021*). We observed signaling by WNT4, WNT5A/B, and WNT6 in both XX and XY gonads as early as E11.5. The number of interactions related to WNT signaling is also increased from E11.5 to E13.5 in both XX and XY gonads even though diminished WNT signaling is required for meiotic entry (*Le Rolle et al., 2021*). However, XY PGCs showed enrichment of the WNT agonist SFRP2 (*van Loon et al., 2021*) at E13.5, indicating that WNT signaling may be inhibited in XY PGCs earlier than XX PGCs. BMP secretion by ovarian supporting cells also plays a critical role in regulating meiotic entry (*Spiller and Bowles, 2022*). Even though BMP6 signaling was observed in XY PGCs, BMP2 and BMP5 signaling were specific to XX PGCs beginning at E11.5. Our findings are consistent with functional studies showing that BMP2 signaling induces *Zglp1* expression and promotes the oogenic fate (*Nagaoka et al., 2020*). Additionally, RA produced by both ovarian and mesonephric somatic cells is required to initiate expression of *Stra8* and *Meiosin*, two TFs that are essential for completion of meiosis (*Spiller and Bowles, 2022*; *Ishiguro et al., 2020*). Significant enrichment of the RA signaling pathway was not detected between supporting cells and PGCs in XX or XY gonads at any embryonic stage with CellphoneDB. Given that RA is primarily produced by the mesonephros (*Bowles et al., 2018*), detection of RA signaling between gonadal supporting cells and PGCs is unlikely. However, *in silico* chromatin binding of RA receptors was enriched in XX PGCs and the RA receptor motifs were found in the regulatory loci for *Rec8, Stra8, Sycp2, Sycp3,* and *Zglp1*, which is consistent with previous studies (*Koubova et al., 2014*). Taken together, our findings provide insight into the temporal patterns of WNT, BMP, and RA signaling during PGC differentiation.

## Discovery analysis identifies potentially new signaling pathways between gonadal supporting cells and PGCs

In addition to these pathways known for their roles in PGC sex determination, we identified several uncharacterized ligand-receptor pairs with sex-specific enrichment. For example, midkine (MDK) signaling was enriched in E13.5 XX gonads. Although the role of MDK signaling in PGCs is not understood, MDK is known to be important for maintaining cell survival and growth and promoting cell migration (*Filippou et al., 2020*). MDK is a heparin binding cytokine and, together with pleiotrophin, regulates female reproductive activities such as oocyte maturation and the estrous cycle (*Muramatsu et al., 2006*). Furthermore, in E13.5 XY gonads, we found enrichment of the Desert Hedgehog (DHH) signaling pathway between *Sox9+* cells and XY PGCs. In addition to regulating fetal Leydig cell differentiation (*Yao et al., 2002*), DHH signaling has also been shown to regulate germ cell maturation and spermatogenesis (*Bitgood et al., 1996*). Ablation of *Dhh* in adult XY mice causes infertility due to a loss of mature sperm (*Bitgood et al., 1996*), however, whether *Dhh* regulates germ cell development in fetal XY mice remains to be determined. Overall, we posit that these and other signaling pathways are important for promoting PGC survival and differentiation.

Our study demonstrated the benefit of using multiomics approaches to probe the cellular and developmental pathways involved in cell fate decisions, gametogenesis, and PGC differentiation. Unlike traditional assays, joint RNA- and ATAC-seq analyses from individual PGCs enabled the detection of

paired chromatin accessibility and gene expression information from the same cell to discover new gene regulatory interactions. We show that our multiomics data accurately identified enrichment of known pathways involved in PGC sex determination, suggesting that our computational approaches are valid and capable of discovering potentially new regulators of PGC fate. Indeed, our findings revealed the diversity of factors required for PGC programming. For example, we cataloged the heterogeneity of PGC populations based on chromatin status and transcription, identified chromatin priming as a potential regulatory mechanism in PGC lineage commitment, and revealed a core set of TFs and ligand-receptor pairs unique to XX and XY PGCs. Taken together, our data demonstrate the power of explorative single-nucleus ATAC-seq and RNA-seq to understand the mechanisms inherent to germline differentiation.

## Materials and methods

### Animals

The single-nucleus multiomics experiments described herein used homozygous Rosa-tdTomato9 females (JAX 007909) (B6.Cg-Gt(ROSA)26Sor<tm9(CAG-tdTomato)Hze>/J) crossed to homozygous Nr5a1-cre mice (B6D2-Tg(Nr5a1-cre)2Klp), provided by the late Dr. Keith Parker (*Bingham et al., 2006*). The immunofluorescence experiments described herein used mice on the C57Bl/6 and CD1 backgrounds. Female mice were timed-mated and the day of detection of vaginal plug was considered embryonic day (E) E0.5. Mice were housed on a 12 hr light:dark cycle, temperature range 70–74°F, and relative humidity range from 40% to 50%. All animal studies were conducted in accordance with the NIH Guide for the Care and Use of Laboratory Animals and approved by the National Institute of Environmental Health Sciences (NIEHS) Animal Care and Use Committee.

### Single-nucleus RNA-seq and ATAC-seq

Whole ovaries and testes were collected at E11.5, E12.5, and E13.5. Plug date was used to determine the stage of embryos collected for single-nucleus RNA-seq and ATAC-seq. The stage of E11.5 embryos was confirmed by counting somites. The stage of embryos collected at E12.5 was confirmed by the morphological presence of the vessel and cords of the testes collected from XY embryos. Similarly, we confirmed the stage of embryos collected at E13.5 by the size of the gonads, the presence of more distinct cords in the testes of XY embryos, and the elongation of the ovaries of XX embryos. The sexes of the embryos were determined using amnion staining, as previously published (*Capel and Batchvarov, 2008*), to identify the presence (XX) or absence (XY) of the Barr body and confirmed with PCR using primers that recognize the Y chromosome. Tomato fluorescence was used to facilitate the dissection of the gonad away from the mesonephros, which was done using needles. Isolated fetal mouse gonads were isolated in PBS, and pairs of gonads or groups pooled between fetuses of the same sex and age were immediately frozen in liquid nitrogen for storage. Nuclei isolation was performed using the 10x Genomics Chromium Nuclei Isolation Kit (cat. # 1000494) following the manufacturer's protocol. The total numbers of gonads used per sex and age are listed in *Supplementary file 1*. The 10x Genomics Chromium Next GEM Single Cell Multiome ATAC+Gene Expression Library Preparation Kit (cat. # 1000284) was used to generate a 10x barcoded library of mRNA and transposed DNA from individual nuclei. The 10x barcoded single-nuclei RNA and ATAC-seq libraries were obtained from 8687 average (1555–20,000) nuclei per sample and sequenced with Illumina NOVAseq to a minimum sequencing depth of 129,120,442 raw reads and 22,406 read pairs/nucleus (*Supplementary file 1*). Two independent technical replicates of multiome sequencing were performed for each sex by embryonic stage combination. The two independent technical replicates comprised of different pools of paired gonads.

### Data preprocessing for single-nucleus RNA-seq and ATAC-seq libraries

*CellRanger* (*Zheng et al., 2017*; *Satpathy et al., 2019*) (v. 3.0) was used for count, alignment, filtering, and cell barcode and UMI counting (*Supplementary file 1*). Barcode swapping correction was performed for all libraries (*Griffiths et al., 2018*). FASTQ files of the corrected cell count matrices were generated and the data were analyzed using *Seurat* (*Satija et al., 2015*) (v. 4.3.0) and *Signac* (*Stuart et al., 2021*) (v. 1.6.0) on R (v. 4.1.0).

## snRNA-seq data preprocessing

Doublets within each dataset were removed with the *scDblFinder* (*Germain et al., 2021*) (v. 1.8.0) package using default methods. Ambient RNA was removed using the *celda/decontX* (*Yang et al., 2020*) (v. 1.12.0) package, with the maximum iterations of the EM algorithm set at 100. Datasets of gonadal cell populations from each embryonic stage and sex were combined into one *Seurat* object using the 'merge' function. As batch effect was not observed in our dataset, we did not perform further integration or batch correction. Cells with nCount_RNA>1000 & nCount_RNA<25,000 & percent. mt<25 were retained for downstream analyses (*Appendix 1—figure 1a*). Mitochondrial genes and transcripts were removed from the snRNA-seq datasets to eliminate any potentially contaminating signal. The data was normalized using the 'SCTransform' function in *Seurat*.

## snATAC-seq data preprocessing

snATAC-seq peak calls from *CellRanger* were used to generate one combined peak set and cells with peak size >20 bp & peak size <10,000 bp were retained for downstream analyses. Fragment objects were created for each library, combined with the respective *Seurat* object, and analyzed using the *Signac* package. A subset of the snATAC-seq data was generated using the following cutoffs: nCount_ATAC<100,000 & nCount_ATAC>1000 & nucleosome_signal<2 & TSS.enrichment>1 (*Appendix 1—figure 1b–d*). Analysis of the snATAC-seq peak feature distribution demonstrated that most peaks occurred in distal intergenic or promoter regions of the genome (*Appendix 1—figure 1e*).

## Sex filtering

To ensure that the correct sex was analyzed in all downstream analyses, we applied computational filters to confirm the sex of the cells in each dataset. Cells were filtered based on Y-linked gene expression and fragments mapped to the Y chromosome (chrY). The *UCell* package (*Andreatta and Carmona, 2021*) (v. 2.0.1) was used to create a module score of the Y-linked genes *Kdm5d, Eif2s3y, Uty,* and *Ddx3y*. Module scores of E11.5 and E12.5 cells above one standard deviation of the mean module score of all XY E13.5 cells were determined to be chromosomally male. Module scores of E11.5 and E12.5 cells below one standard deviation of the mean module score of all XX E13.5 cells were determined to be chromosomally female. Next, we examined the peaks called on chrY outside of the pseudoautosomal region. In total, 21 peaks were identified between chrY 1–90,000,000 bp. Cells with 0 fragment counts within our defined chrY peak region or chrY gene expression module score<1 standard deviation were removed from the male datasets. Cells with >1 fragment counts within our defined chrY peak region or chrY gene expression module score >1 standard deviation were removed from the female datasets.

## Cell-type identification

To identify the cellular origin of the snRNA-seq clusters, principal component analysis (PCA) was run on the scaled data, and visualization was performed using UMAP with dims = 18 and resolution = 0.3. PGC cluster identification was performed using the established germ cell markers *Ddx4* and *Pou5f1* (*van den Bergen et al., 2009*). To remove potential PGC and gonadal somatic cell doublets, only cells with RNA count<1 for the following genes were retained as PGCs: the granulosa cell marker genes *Foxl2* and *Runx1* (*Stévant et al., 2019*); Sertoli cell marker *Sox9* (*Stévant et al., 2019*); Leydig cell marker *Insl3* (*Stévant et al., 2019*); stromal cell marker *Wt1* (*Zhao et al., 2009*); endothelial cell marker *Plvap* (*Denzer et al., 2023*); immune-related cell marker *Mafb* (*Kelly et al., 2000*); interstitial cell markers *Pdgfra* and *Nr2f2* (*Stévant et al., 2019*); supporting cell marker *Tspan8* (*Garcia-Alonso et al., 2022*); and epithelial cell marker *Krt19* (*Kuony and Michon, 2017*). Somatic supporting cell cluster identification was performed using the established female and male support cell markers *Runx1* and *Sox9*, respectively. To remove potential gonadal supporting cell doublets formed with germ cells or interstitial cells, only cells with RNA count>1 for *Wt1* and RNA count<1 for the following genes were retained as support cells: the germ cell marker genes *Ddx4* and *Pou5f1*; endothelial cell markers *Plvap* and *Pecam1* (*Watt et al., 1995*); and interstitial cell markers *Pdgfra* and *Nr2f2*.

## Clustering and pseudotime trajectory of PGCs

snRNA-seq reclustering of PGCs was done with PCA on scaled PGC RNA data, and visualization was performed using UMAP with dimensions (dims)=18 and resolution = 0.3. snATAC-seq clustering of

PGCs was done first by calling peaks using *MACS2* (*Zhang et al., 2008*) (v. 2.2.7.1). Peaks on nonstandard chromosomes were removed using the 'keepStandardChromosomes' function in *Signac* with the option pruning.mode = 'coarse', and peaks in mm10 genomic blacklist regions were removed. Counts in each peak were determined using *Signac* functionalities. The data was normalized by latent semantic indexing (LSI) using the 'RunTFIDF', 'FindTopFeatures', and 'RunSVD' functions in *Signac*. Visualization was performed using UMAP with dims = 2:30, reduction = 'lsi', and algorithm = 3. The joint neighbor graph or joint UMAP was generated using the 'FindMultiModalNeighbors' function in Signac and reduction.name = 'wnn.umap'. A Seurat object from the clustered PGCs was used as input for pseudotime analysis with Monocle3 to determine the differentiation trajectory of PGCs using both female and male E11.5 PGCs as time '0' in pseudotime.

### Identification of DEGs and DAPs

DEGs in each cluster were determined using the 'FindAllMarkers' function in *Seurat* based on the Wilcoxon rank-sum test. DEGs between female and male PGCs at each embryonic stage were identified with the 'FindMarkers' function in *Seurat* with min.pct=0.25 and logfc.threshold=0.25. DAPs between female and male PGCs were identified with the 'FindMarkers' function in *Signac* with min.pct=0.001 and logfc.threshold=0.1.

### Correlation of peak accessibility with gene expression

snATAC-seq peaks were linked to gene expression, as imputed from the snRNA-seq data, using the 'LinkPeaks' function in *Signac*. Peaks linked to DEGs for each sex and embryonic stage were then subsetted out for further downstream analyses. Plots showing peak-to-gene linkages were generated using the 'CoveragePlot' function in *Signac*.

### Motif enrichment analysis of TF binding sites

DNA sequence motif analysis of peaks linked to DEGs was performed using the 'FindMotifs' function in *Signac*. Motifs were obtained from the JASPAR 2020 database (collection = 'Core' and taxonomic group = 'vertebrates'). Background peaks were selected to match the GC content in the peak set by using the 'AccessiblePeaks', 'GetAssayData', and 'MatchRegionStats' functions in *Signac*. *MotifScan* (*Sun et al., 2018*) (v. 1.3.0) was used to determine the genomic position of linked peaks to DEGs to identify predicted target genes of TFs. Per-cell motif activity was computed using the 'RunChromVAR' function in *Signac*.

### Module score calculation for gene expression

Module scores of gene expression were calculated using the 'AddModuleScore' function in *Seurat*. Module scores were plotted on UMAPs using the 'FeaturePlot' function in *Seurat*.

### Peak feature distribution

Pie charts of peak feature distribution were generated with the R package *ChIPseeker* (*Wang et al., 2022*; *Yu et al., 2015a*) (v. 1.38.0). Peaks were annotated with the 'annotatePeak' function with options 'tssRegion = c(–3000,3000)', 'TxDb = TxDb.Mmusculus.UCSC.mm10.knownGene', and 'annoDb=org. Mm.eg.db'. Pie charts were plotted with the function 'plotAnnoPie'.

### GO annotation and enrichment

The GO enrichment analyses were carried out using the R package *clusterProfiler* (*Wu et al., 2021*) (v. 4.0.5) with the 'enrichGO' function. The Benjamini-Hochberg method (a=0.05) was used to control for FDR. GO annotation enrichment for biological processes were plotted using the 'dotplot' or 'cnetplot' function of the *Enrichplot* (*Yu, 2021*) (v. 1.12.2) R package.

### Cistrome DB Regulatory Potential Score

The Cistrome DB Regulatory Potential Score (*Zheng et al., 2019*; *Mei et al., 2017*) was used to identify potential regulators of *Bnc2* and *Stra8*. The toolkit (http://dbtoolkit.cistrome.org) was used for these analyses by selecting Mouse mm10 as 'Species', TF/chromatin regulator as 'Data Type in Cistrome', and 10 kb as 'The half-decay distance to transcription start site'.

## Enrichment of signaling pathways between gonadal support cells and PGCs

*CellphoneDB* (*Troulé, 2023*) (v. 5.0.0) was used to predict paired ligand-receptor pairs for cell-cell communication between gonadal supporting cells and PGCs. Normalized snRNA-seq counts were used as input data. Significant ligand-receptor pairs were determined independently between gonadal supporting cells and PGCs for each embryonic stage and sex. The statistical analysis (method 2) of *CellphoneDB* and the 'statistical_analysis_method' function in *CellphoneDB* were used to identify significantly enriched signaling pathways.

### Immunofluorescence

Gonads were collected from E13.5 XX and XY embryos and fixed in 4% PFA for 1 hr at room temperature. Gonads were then washed in PBS and stored at 4°C until embedding in paraffin and sectioned at 5 μm thickness. Following deparaffinization, sections were rehydrated in decreasing alcohol concentration gradients. Antigen retrieval was performed with 0.1 mM citric acid (Vector Labs) for 20 min in the microwave at 10% power and then cooled to room temperature. Samples were blocked in PBS-Triton X-100, 0.1% solution with 5% normal donkey serum for 1 hr. Samples were incubated with primary antibodies diluted in blocking buffer (PBS-Triton X-100, 0.1% solution with 5% normal donkey serum) at 4°C overnight. Samples were then washed with PBS-Triton X-100, 0.1% solution, and incubated with secondary antibodies diluted in blocking buffer at room temperature for 1 hr. The slides were washed with PBS-Triton X-100, 0.1% solution 3× and incubated with the Vector TrueView Autofluorescence Quenching Kit (cat. # SP-8400) for 2 min. Following incubation with the quenching kit, the slides were washed with PBS-Triton X-100, 0.1% solution 2×, PBS 1×, and incubated with DAPI (Invitrogen cat. # D1306). The slides were then mounted with slide mounting media (Vectashield Vibrance Antifade Mounting Medium TrueView Kit SP-8400). Primary antibodies used included: Anti-TRA98 (1:500; rat, BioAcademia cat. # 73-003), Anti-NR2F2 (1:300, mouse, R&D Systems cat. # PP-H7147-00), Anti-PORCN (1:200, rabbit, Thermo Fisher cat. # PA5-43423), and Anti-AP-2γ (TFAP2c) (1:200, mouse, Santa Cruz cat. # sc-12762). Secondary antibodies were used at 1:200 dilution (Invitrogen/Life Technology): Donkey anti-rat Alexa 568 (A78946), Donkey anti-mouse Alexa 647 (A31571), and Donkey anti-rabbit Alexa 488 (A21206). Imaging for sections was performed on a Zeiss LSM 900 confocal microscope using Zen software. Brightness and contrast of images were adjusted using ImageJ (National Institutes of Health, USA).

### Statistical analyses

Libraries were prepared from 3 to 19 pairs of gonads and two technical replicates per embryonic stage and sex (*Supplementary file 1*). Statistical analyses were considered significant if p<0.05.

## Acknowledgements

We thank the late Dr. Keith Parker (UT Southwestern Medical Center, USA) for the Nr5a1-cre mice. Valuable discussions with F DeMayo, C Williams, M Morgan, C Guardia, and all members of the Yao lab during the course of this project contributed greatly to our manuscript. We are especially grateful to P Brown for her advice and assistance with experiments and editorial comments in the writing process. We are grateful to the NIEHS Comparative Medicine Branch for mouse colony maintenance and to the Epigenomics and DNA Sequencing Core and the Integrative Bioinformatics Support Group for their assistance with sequencing and data analysis. This work was supported by the Intramural Research Program (ES102965 to HH-CY) of the NIH, National Institute of Environmental Health Sciences and the Postdoctoral Research Associate Training (PRAT) Program (Fi2) Fellowship of the NIH, National Institute of General Medical Sciences awarded to AKA.

# Additional information

## Funding

| Funder | Grant reference number | Author |
| --- | --- | --- |
| National Institute of Environmental Health Sciences | ZIAES102965 | Humphrey HC Yao |

The funders had no role in study design, data collection and interpretation, or the decision to submit the work for publication.

## Author contributions

Adriana K Alexander, Conceptualization, Data curation, Formal analysis, Validation, Investigation, Methodology, Writing – original draft, Writing – review and editing; Karina F Rodriguez, Yu-Ying Chen, Ciro Amato, Data curation, Methodology, Writing – review and editing; Martin A Estermann, Barbara Nicol, Investigation, Writing – review and editing; Xin Xu, Data curation, Writing – review and editing; Humphrey HC Yao, Conceptualization, Formal analysis, Supervision, Funding acquisition, Project administration, Writing – review and editing

## Author ORCIDs

Adriana K Alexander (b) https://orcid.org/0000-0003-1389-0228
Martin A Estermann (b) https://orcid.org/0000-0002-8623-2720
Humphrey HC Yao (b) https://orcid.org/0000-0003-2944-8469

Reviewer #1 (Public review): https://doi.org/10.7554/eLife.96591.3.sa1
Reviewer #2 (Public review): https://doi.org/10.7554/eLife.96591.3.sa2
Reviewer #3 (Public review): https://doi.org/10.7554/eLife.96591.3.sa3
Author response https://doi.org/10.7554/eLife.96591.3.sa4

# Additional files

## Supplementary files

Supplementary file 1. Sequencing and QC statistics for single-nucleus multiome libraries of E11.5-E13.5 XX and XY gonads.

MDAR checklist

## Data availability

Sequencing data have been deposited in GEO under accession code GSE288206.

The following dataset was generated:

| Author(s) | Year | Dataset title | Dataset URL | Database and Identifier |
| --- | --- | --- | --- | --- |
| Alexander AK, Rodriguez KF, Chen YY, Amato CM, Estermann MA, Nicol B, Xu X | 2025 | Single-nucleus multiomics reveals the gene-reguatory networks underlying sex determination of murine primordial germ cells | https://www.ncbi.nlm.nih.gov/geo/query/acc.cgi?acc=GSE288206 | NCBI Gene Expression Omnibus, GSE288206 |

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

## Appendix 1

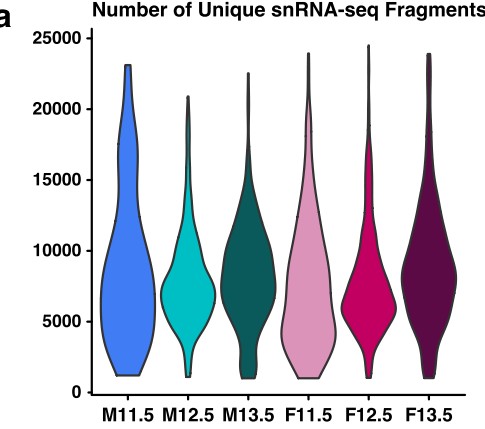

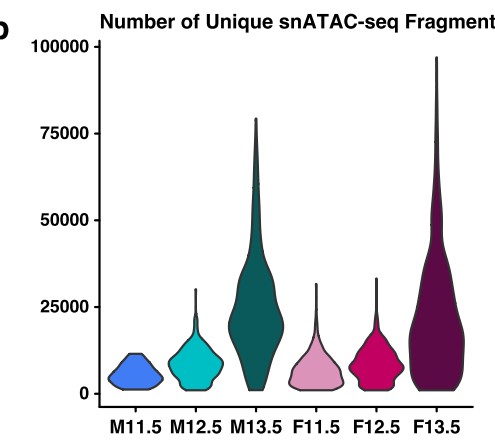

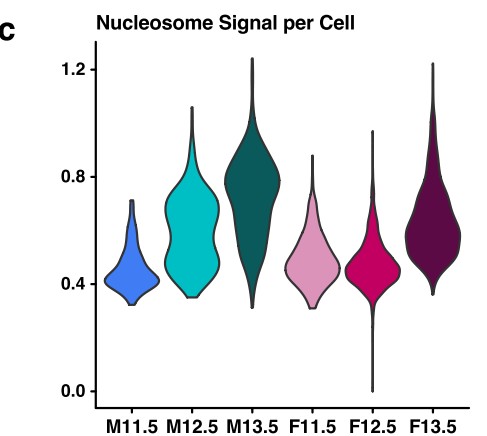

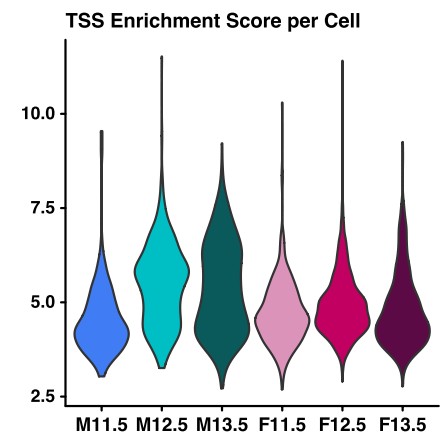

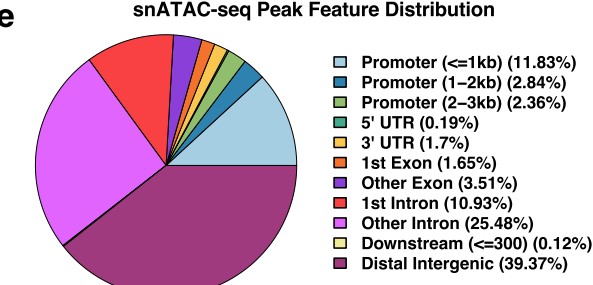

**Appendix 1—figure 1.** QC statistics for the snRNA-seq and snATAC-seq libraries of E11.5-E13.5 XX and XY primordial germ cells (PGCs). (**a**) Violin plot of the number of unique snRNA-seq reads per cell for E11.5-E13.5 XX and XY PGCs. (**b**) Violin plot of the number of unique snATAC-seq fragments per cell for E11.5-E13.5 XX and XY PGCs. (**c**) snATAC-seq nucleosome signal per cell for E11.5-E13.5 XX and XY PGCs. (**d**) snATAC-seq transcription start site (TSS) enrichment score per cell for E11.5-E13.5 XX and XY PGCs. (**e**) Feature distribution of snATAC-seq peaks called for E11.5-E13.5 XX and XY PGCs.



**Appendix 1—figure 2.** Heatmap of marker gene expression for snRNA-seq clusters of E11.5- E13.5 XX and XY primordial germ cells (PGCs). (**a**) Heatmap of marker gene expression for unbiasedly identified snRNA clusters of E11.5-E13.5 XX and XY PGCs. The color scale represents the expression level of each gene. The colored bar at top of heatmap represents the cluster number.

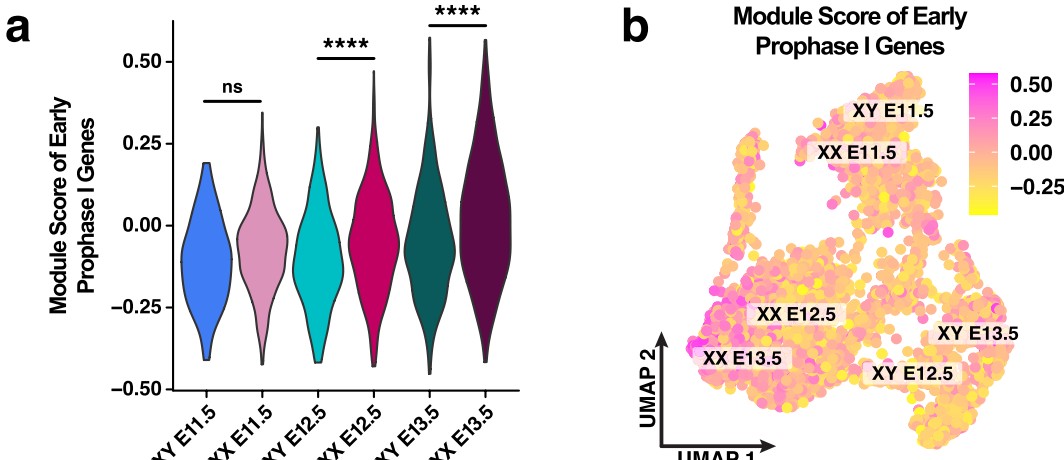

**Appendix 1—figure 3.** Module score of early prophase I genes. (**a**) Violin plot of the module score, or average expression, of early prophase I genes for each primordial germ cell (PGC) population clustered by embryonic stage and sex. ****p<0.0001; one-way ANOVA with post hoc Tukey multiple comparison test. (**b**) Joint Uniform Manifold Approximation and Projection (UMAP) of integrated snRNA-seq and snATAC-seq data showing the module score levels of early prophase I genes across all PGC populations. (**a–b**) Early prophase I genes included Rad21, Rad21l, Rec8, Ccnb3, Sycp2, Rad51, Hormad, Sycp1, and Kit.



**Appendix 1—figure 4.** Regulatory potential of retinoic acid receptors. (**a**) Dotplot of the average expression and percentage of cells expressing Rara, Rarb, Rarg, Rxra, Rxrb, Rxrg, Crabp1, and Crabp2. The color scale represents the average expression level, and the size of the dot represents the percentage of cells expressing the gene. (**b**) Joint Uniform Manifold Approximation and Projection (UMAP) showing Rara, Rarb, Rxrb, and Crabp2 expression levels for E11.5-E13.5 XX and XY primordial germ cells (PGCs). Individual cells are color coded by the level of gene expression. (**c**) In silico chromatin binding score, termed 'motif activity', of retinoic acid receptors. Average motif activity is indicated by the color scale, and the percentage of cells is indicated by the size of the dot. (**d**) Joint UMAP showing the motif activity scores of RARA, RARB, RXRB, and RXRG for E11.5-E13.5 XX and XY PGCs. (**e**) Pie chart showing the feature distribution of the RARB and RXRA motifs. (**f**) Gene ontology (GO) of predicted target genes of RARB and RXRA based on motif-to-gene linkages. The color scale represents the

*Appendix 1—figure 4 continued on next page*

*Appendix 1—figure 4 continued*
p-value of GO term enrichment, and the size of the dot indicates the gene ratio for each GO term. (**g**) Icon array showing the presence (pink) or absence (white) of the retinoic acid (RA) receptor motif (y-axis) in peaks linked to meiotic genes.

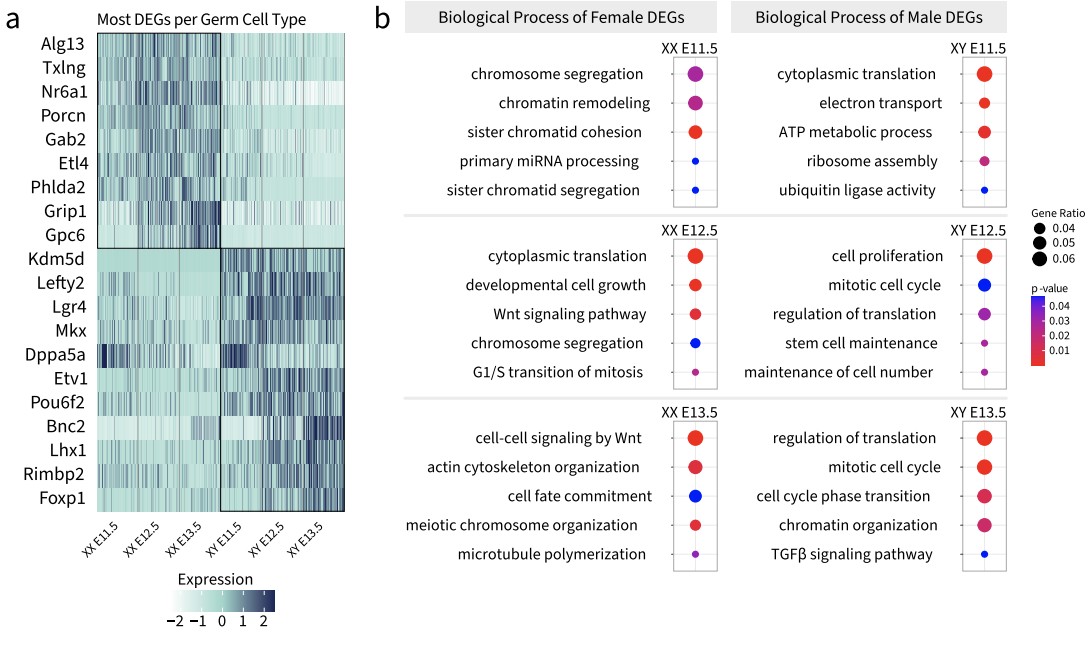

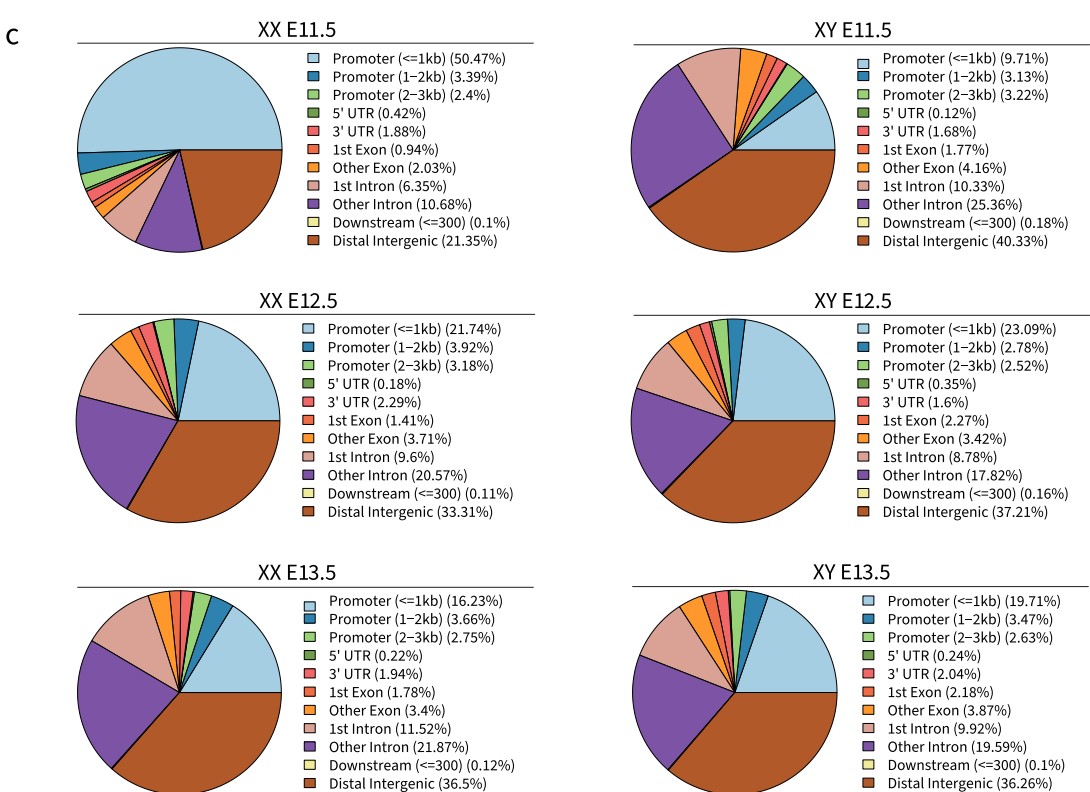

**Appendix 1—figure 5.** Characterization of differentially expressed genes (DEGs) and differentially accessible peaks in E11.5-E13.5 XX and XY primordial germ cells (PGCs). (**a**) Heatmap of the most DEGs for each PGC population. The color scale represents the level of expression. (**b**) Biological process gene ontology (GO) analysis

*Appendix 1—figure 5 continued on next page*

*Appendix 1—figure 5 continued*

of XX and XY DEGs. The color scale represents the p-value of GO term enrichment, and the size of the dot indicates the gene ratio for each GO term. (**c**) Pie charts showing the feature distribution of differentially accessible chromatin peaks for E11.5-E13.5 XX and XY PGCs.

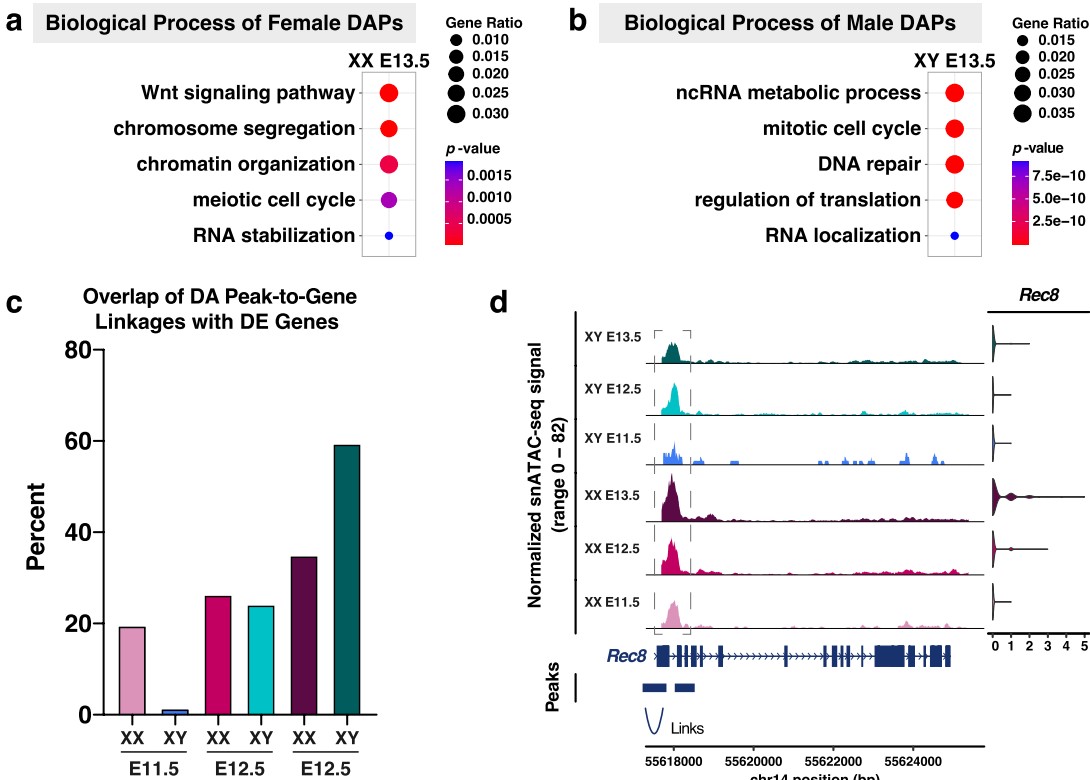

**Appendix 1—figure 6.** Characterization of peak-to-gene linkages in E11.5-E13.5 XX and XY primordial germ cells (PGCs). (**a–b**) Biological process gene ontology (GO) analysis of E13.5 XX (**a**) and XY (**b**) differentially accessible peaks (DAPs). The color scale represents the p-value of GO term enrichment, and the size of the dot indicates the gene ratio for each GO term. (**c**) Bar plot showing the percentage of DAPs with peak-to-gene linkages to differentially expressed genes. (**d**) *Left*: Coverage plot of the normalized snATAC-seq signal at the Rec8 locus. Peak-to-gene linkages are indicated by the 'links' line. *Right*: Violin plot of expression in E11.5-E13.5 XX and XY PGCs.

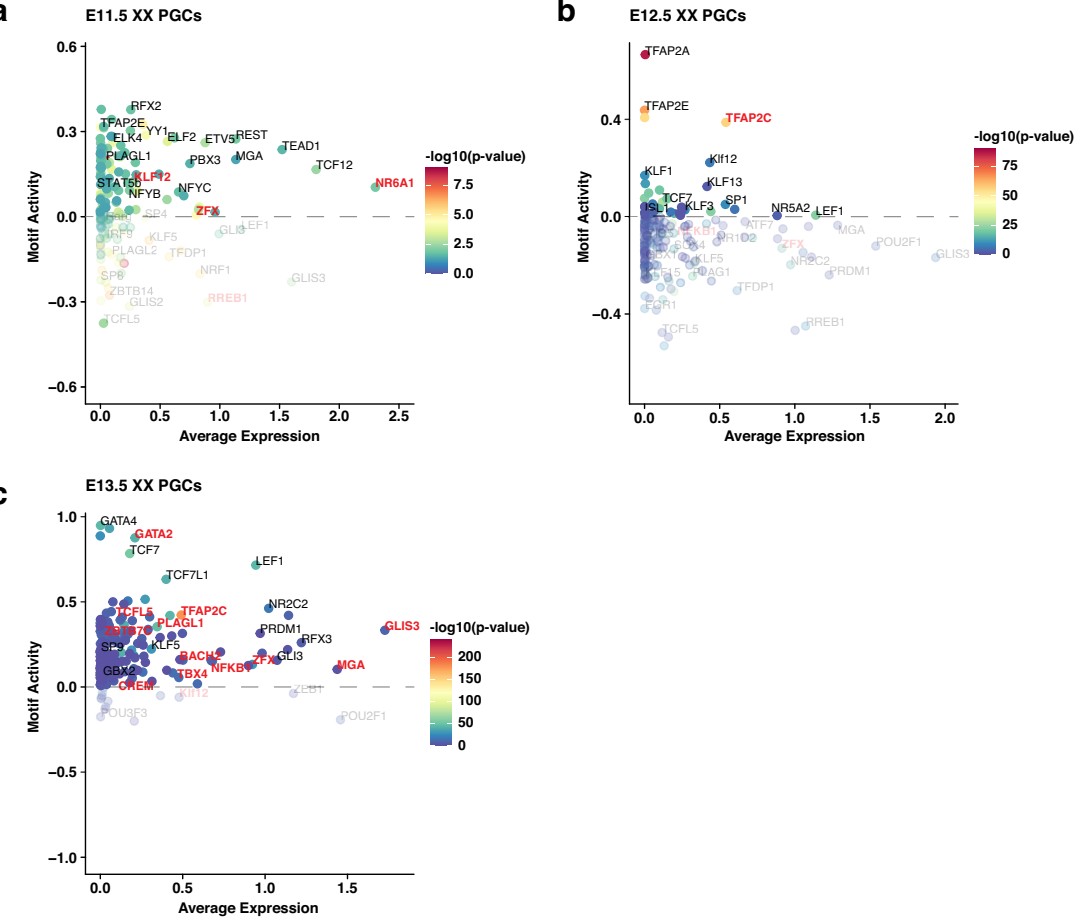

**Appendix 1—figure 7.** Identification of enriched transcription factors (TFs) in XX primordial germ cells (PGCs). (**a–c**) In silico chromatin binding score, termed 'motif activity', vs average expression of TFs that bind significantly enriched motifs in E11.5 (**a**), E12.5 (**b**), and E13.5 (**c**) XX PGCs. The color scale represents the -log10(p-value) of motif enrichment. TF names colored in red are significantly upregulated in XX PGCs when compared to XY PGCs.



**Appendix 1—figure 8.** Identification of strong transcription factor (TF) candidates in XX primordial germ cells (PGCs). (**a**) Dotplot of the average expression and percentage of cells expressing XX PGC-enriched TFs in PGCs and gonadal somatic cells. Dashed boxes indicate TFs that are enriched in PGCs when compared to the somatic compartment. The color scale represents the average expression level, and the size of the dot represents the percentage of cells expressing the gene. (**b–f**) Line plots showing the expression levels and in silico chromatin binding, termed 'motif activity', of Tbx4 (**b**), Nr6a1 (**c**), Gata2 (**d**), Zfx (**d**), and Mga (**f**) in E11.5-E13.5 XX (pink) and XY (teal) PGCs.

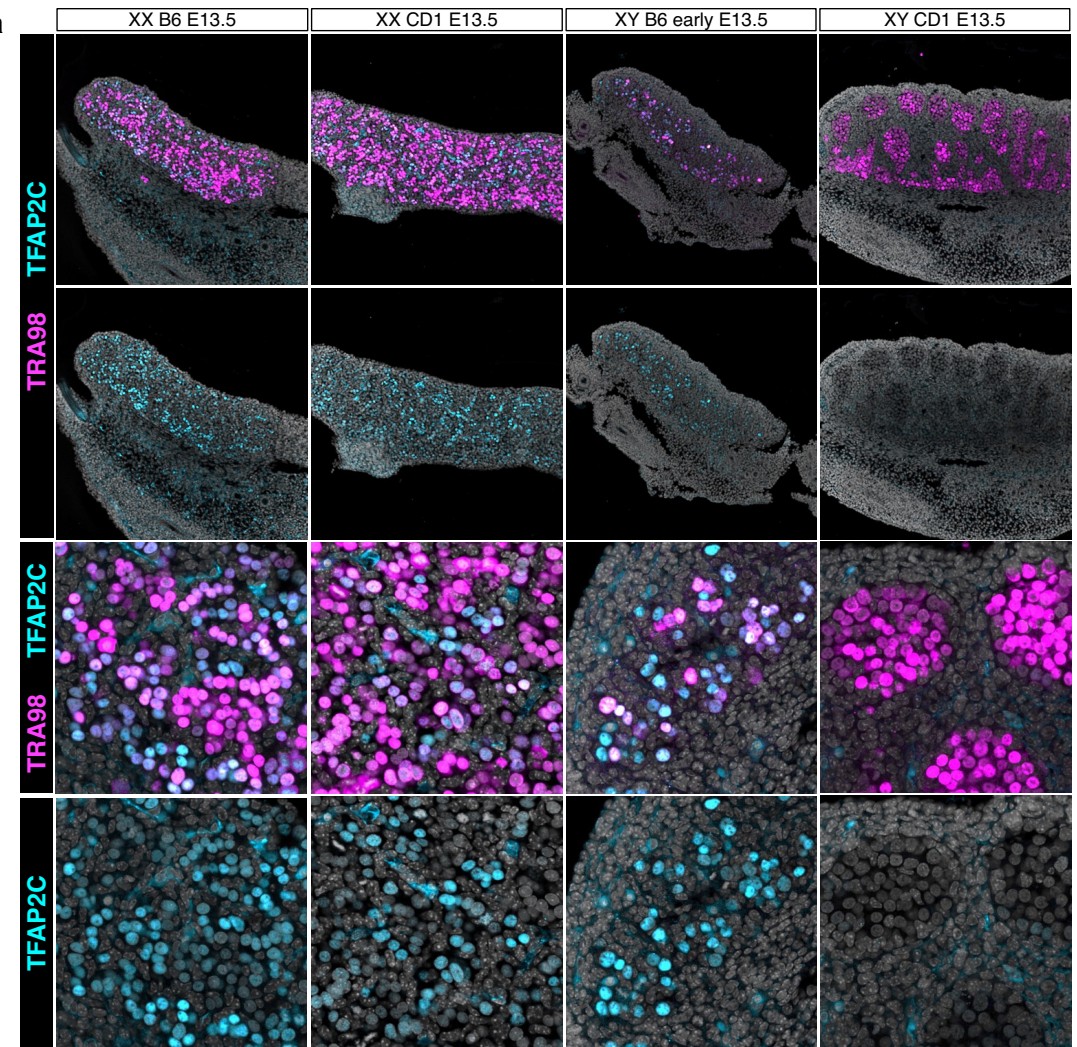

**Appendix 1—figure 9.** Immunofluorescence staining of TFAP2C in E13.5 gonads. (**a**) DAPI staining (gray) and immunofluorescence staining of TRA98 (pink) and TFAP2C (blue) in E13.5 gonads from XX and XY mice on the C57Bl/6 or CD1 background.

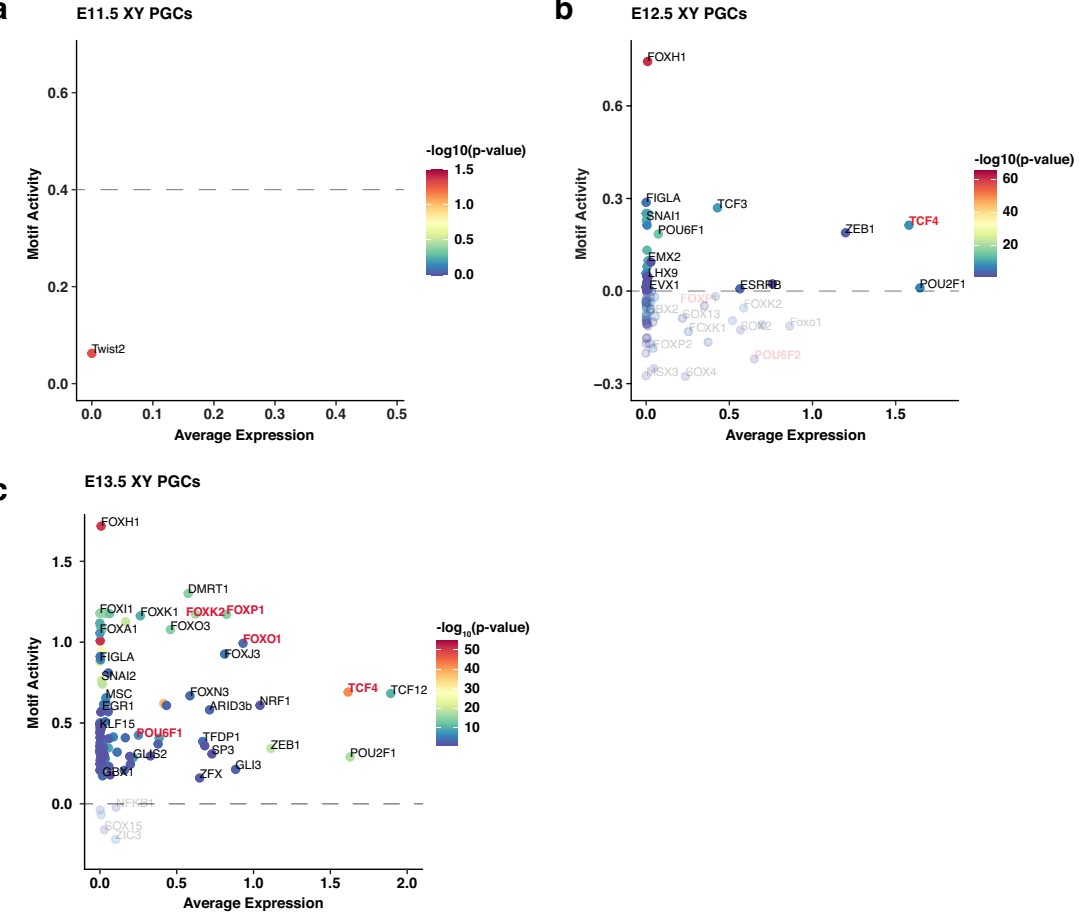

**Appendix 1—figure 10.** Identification of enriched transcription factors (TFs) in XY primordial germ cells (PGCs). (**a–c**) In silico chromatin binding score, termed 'motif activity', vs average expression of TFs that bind significantly enriched motifs in E11.5 (**a**), E12.5 (**b**), and E13.5 (**c**) XY PGCs. The color scale represents the -log10(p-value) of motif enrichment. TF names colored in red are significantly upregulated in XY PGCs when compared to XX PGCs.

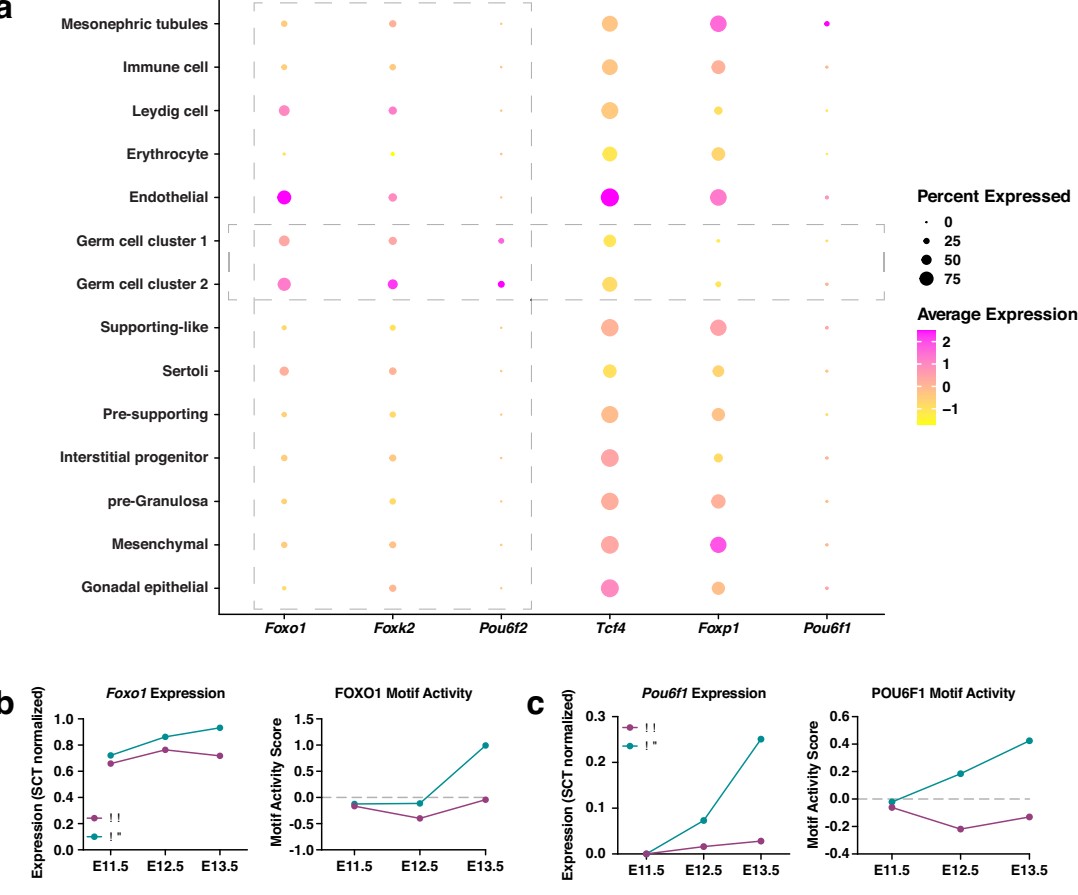

**Appendix 1—figure 11.** Identification of compelling transcription factor (TF) candidates enriched in XY primordial germ cells (PGCs). (**a**) Dotplot of the average expression and percentage of cells expressing XY PGC-enriched TFs in PGCs and gonadal somatic cells. Dashed boxes indicate TFs that are enriched in PGCs when compared to the somatic compartment. The color scale represents the average expression level, and the size of the dot represents the percentage of cells expressing the gene. (**b–c**) Line plots showing the expression levels and in silico chromatin binding, termed 'motif activity', of Foxo1 (**b**) and Pou6f1 (**c**) in E11.5-E13.5 XX (pink) and XY (teal) PGCs.

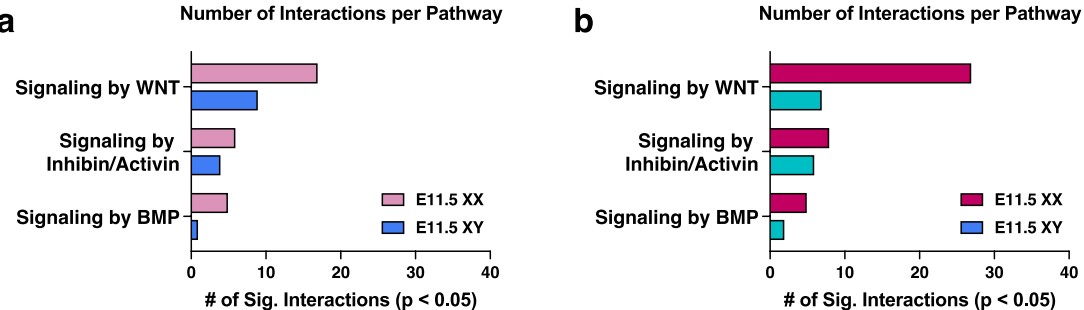

**Appendix 1—figure 12.** Number of interactions per signaling by WNT, inhibin/activin, and BMP pathways in XX and XY gonads. (**a–b**) Number of significantly enriched ligand-receptor pairs, or interactions, per the signaling by WNT, inhibin/activin, and BMP pathways in E11.5 (**a**) and E12.5 (**b**) XX and XY gonads.

