## [Editor Report · eLife Assessment]

This **important** study reports single-nucleus multiomics-based profiling of transcriptome and chromatin accessibility of mouse XX and XY primordial germ cells (PGCs). The main conclusions of this study, which will be of interest to developmental and reproductive biologists, as well as andrologists, are supported by **convincing** data.

---

## [Referee Report · Reviewer #1 (Public review)]

Summary:

This study uses single nucleus multi-omics to profile the transcriptome and chromatin accessibility of mouse XX and XY primordial germ cells (PGCs) at three time points spanning PGC sexual differentiation and entry of XX PGCs into meiosis (embryonic days 11.5-13.5). They find that PGCs can be clustered into sub-populations at each time point, with higher heterogeneity among XX PGCs and more switch-like developmental transitions evident in XY PGCs. In addition, they identify several transcription factors that appear to regulate sex-specific pathways as well as cell-cell communication pathways that may be involved in regulating XX vs XY PGC fate transitions. The findings are important and overall rigorous. The study could be further improved by better connection to the biological system, including putting the transcriptional heterogeneity of XX PGCs in the context of findings that meiotic entry is spatially asynchronous in the fetal ovary and further addressing the role of retinoic acid signaling. Overall, this study represents and advance in germ cell regulatory biology and will be a highly used resource in the field of germ cell development.

Strengths:

(1) The multi-omics data is mostly rigorously collected and carefully interpreted.

(2) The dataset is extremely valuable and helps to answer many long-standing questions in the field.

(3) In general, the conclusions are well anchored in the biology of the germ line in mammals.

Comments on revised version:

Most of my concerns have been addressed in the revised manuscript. I have one remaining concern but I believe this is important in order for the paper to be fully appreciated:

In Figures 2a, 2e, 3a, and 3e, the visualization scheme is very difficult to follow, and has not been updated or improved in the revised manuscript. It's very hard to see the colors corresponding to average expression for many genes because the circles are so small. The yellow color is hard to see and makes it hard to estimate the size of the circle. This issue is particularly egregious in Figure 2a for the data relating to ZKSCAN5, which is specifically highlighted in the text in lines 421-426. This data must be shown in a more convincing way in order to make the claims. An update to the visualization, including color scheme, is very strongly recommended; it is not difficult and would substantially improve the ability of these panels to communicate their message.

---

## [Referee Report · Reviewer #2 (Public review)]

Summary:

This manuscript by Alexander et al describes a careful and rigorous application of multiomics to mouse primordial germ cells (PGCs) and their surrounding gonadal cells during the period of sex differentiation.

Strengths:

In thoughtfully designed figures, the authors identify both known and new candidate gene regulatory networks in differentiating XX and XY PGCs and sex-specific interactions of PGCs with supporting cells. In XY germ cells, novel findings include the predicted set of TFs regulating Bnc2, which is known to promote mitotic arrest, as well as the TFs POU6F1/2 and FOXK2 and their predicted targets that function in mitosis and signal transduction. In XX germ cells, the authors deconstruct the regulation of the premeiotic replication factor Stra8, which reveals TFs involved in meiosis, retinoic acid signaling, pluripotency and epigenetics among predictions; this finding, along with evidence supporting regulatory potential of retinoic acid receptors in meiotic gene expression is an important addition to the debate over the necessity of retinoic acid in XX meiotic initiation. In addition, a self-regulatory network of other TFs is hypothesized in XX differentiating PGCs, including TFAP2c, TCF5, ZFX, MGA and NR6A1, which is predicted to turn on meiotic and Wnt signaling targets. Finally, analysis of PGC-support cell interactions during sex differentiation reveals substantially more interactions in XX, via WNTs and BMPs, as well as some new signaling pathways that predominate in XY PGCs including ephrins, CADM1, Desert Hedgehog and matrix metalloproteases. This dataset will be an excellent resource for the community, motivating functional studies and serving as a discovery platform.

Weaknesses:

While the authors performed all of their comparisons between XX versus XY datasets at each timepoint, a more systematic analysis of expression and accessibility changes across time for each sex would be valuable. It remains possible that common mechanisms of differentiation to XX and XY could be missing from this analysis that focused on sex-specific differences.

Specific Questions:

(1) Line 461: "the population of E13.5 XX PGCs displaying the strongest Stra8 expression levels corresponded to the same population of XX PGCs with the highest module score of early meiotic prophase I genes (Fig. 3c; Supplementary Fig. 3a-b)" however the Stra8+ XX PGCs that do not robustly express meiotic genes should be examined to understand more about their differentiation potential. The authors are well-poised to identify the likely trajectories available to cell subsets in their dataset, and not doing so is a missed opportunity.

(2) The authors state that "we found that Stra8, Rec8, Rnf2, Sycp1, Sycp2, Ccnb3, and Zglp1 contain the RA receptor motifs in their regulatory sequences (Supplementary Figure 4g)." What is the strength of the RA->meiosis pathway compared to other mechanisms regulating meiosis? Perhaps the authors could take this analysis further with the following questions: (1) ask whether meiotic genes more enriched in RA motifs compared to other expressed genes or other motifs (2) compare the strength of peak-gene correlations for all peaks containing RA receptor motifs vs. those with peaks for Zglp1, Rnf2, etc binding. The strengths of these correlations could provide clues to how much gene expression varies in response to RA exposure vs. modulation of these other factors and thus tell us something about how much RA is playing a role.

(3) In figure 4, the shift from promoters in E11.5 XX PGCs to distal intergenic regions is fascinating. What can we learn about epigenetic reprogramming/methylation changes across gene bodies?

(4) The overlap between gene targets of TCFL5 with other highly expressed TFs differentially upregulated in E13.5 XX PGCs over XY suggests ambiguity regarding its role as a central or high-level regulator of differentiation; as in vivo validation has not been performed, I suggest softening this conclusion.

---

## [Referee Report · Reviewer #3 (Public review)]

Summary:

Alexander et al. reported the gene-regulatory networks underpinning sex determination of murine primordial germ cells (PGCs) through single-nucleus multiomics, offering a detailed chromatin accessibility and gene expression map across three embryonic stages in both male (XY) and female (XX) mice. It highlights how regulatory element accessibility may precede gene expression, pointing to chromatin accessibility as a primer for lineage commitment before differentiation. Sexual dimorphism in these elements and gene expression increases over time, and the study maps transcription factors regulating sexually dimorphic genes in PGCs, identifying sex-specific enrichment in various transcription factors.

Strengths:

The study includes step-wise multiomic analysis with some computational approach to identify candidate TFs regulating XX and XY PGC gene expression, providing a detailed timeline of chromatin accessibility and gene expression during PGC development, which identifies previously unknown PGC subpopulations and offers a multimodal reference atlas of differentiating PGC clusters. Furthermore, the study maps a complex network of transcription factors associated with sex determination in PGCs, adding depth to our understanding of these processes.

Weaknesses:

While the multiomics approach is powerful, it primarily offers correlational insights between chromatin accessibility, gene expression, and transcription factor activity, without direct functional validation of identified regulatory networks.

Comments on revised version:

The authors have answered my questions and concerns in the revised manuscript and correspondence.

---

## [Author Response]

The following is the authors’ response to the original reviews.

**Public Reviews:**

**Reviewer #1 (Public Review):**
Summary:This study uses single nucleus multiomics to profile the transcriptome and chromatin accessibility of mouse XX and XY primordial germ cells (PGCs) at three time-points spanning PGC sexual differentiation and entry of XX PGCs into meiosis (embryonic days 11.5-13.5). They find that PGCs can be clustered into sub-populations at each time point, with higher heterogeneity among XX PGCs and more switch-like developmental transitions evident in XY PGCs. In addition, they identify several transcription factors that appear to regulate sex-specific pathways as well as cell-cell communication pathways that may be involved in regulating XX vs XY PGC fate transitions. The findings are important and overall rigorous. The study could be further improved by a better connection to the biological system, including the addition of experiments to validate the 'omics-based findings in vivo and putting the transcriptional heterogeneity of XX PGCs in the context of findings that meiotic entry is spatially asynchronous in the fetal ovary. Overall, this study represents an advance in germ cell regulatory biology and will be a highly used resource in the field of germ cell development.Strengths:(1) The multiomics data is mostly rigorously collected and carefully interpreted.(2) The dataset is extremely valuable and helps to answer many long-standing questions in the field.(3) In general, the conclusions are well anchored in the biology of the germ line in mammals.Weaknesses:(1) The nature of replicates in the data and how they are used in the analysis are not clearly presented in the main text or methods. To interpret the results, it is important to know how replicates were designed and how they were used. Two "technical" replicates are cited but it is not clear what this means.

The two independent technical replicates comprised different pools of paired gonads. This sentence was added to the methods section of the revised manuscript.

(2) Transcriptional heterogeneity among XX PGCs is mentioned several times (e.g., lines 321-323) and is a major conclusion of the paper. It has been known for a long time that XX PGCs initiate meiosis in an anterior-to-posterior wave in the fetal ovary starting around E13.5. Some heterogeneity in the XX PGC populations could be explained by spatial position in the ovary without having to invoke novel subpopulations.

We thank the reviewer for pointing out this important biological phenomenon. We also recognize that transcriptional heterogeneity among XX PGCs is likely due to the anterior-to-posterior wave of meiotic initiation in E13.5 ovaries and highlight this possibility in our manuscript. However, since our study utilizes single-nucleus RNA-sequencing and not spatial transcriptomics, we are not able to capture the spatial location of the XX PGCs analyzed in our dataset. As such, our analysis applied clustering tools to classify the populations of XX PGCs captured in our dataset.

(3) There is essentially no validation of any of the conclusions. Heterogeneity in the expression of a given marker could be assessed by immunofluorescence or RNAscope.

In our revised manuscript, we included immunofluorescence staining of potential candidate factors involved in PGC sex determination, such as PORCN and TFAP2C. Testing and optimizing antibodies for the targets identified in this study are ongoing efforts in our lab and we look forward to sharing our results with the research community.

(4) The paper sometimes suffers from a problem common to large resource papers, which is that the discussion of specific genes or pathways seems incomplete. An example here is from the analysis of the regulation of the Bnc2 locus, which seems superficial. Relatedly, although many genes and pathways are nominated for important PGC functions, there is no strong major conclusion from the paper overall.

In this manuscript, we set out to identify candidate factors, some already known and many others unknown, involved in the developmental pathways of PGC sex determination using computational tools. Our goal, as a research group and with future collaborators, is to screen these interesting candidates and discover their function in the primordial germ cell. Our research, presented in this study, represents a launching pad for which to identify future projects that will investigate these factors in further detail.

**Reviewer #2 (Public Review):**
Summary:This manuscript by Alexander et al describes a careful and rigorous application of multiomics to mouse primordial germ cells (PGCs) and their surrounding gonadal cells during the period of sex differentiation.Strengths:In thoughtfully designed figures, the authors identify both known and new candidate gene regulatory networks in differentiating XX and XY PGCs and sex-specific interactions of PGCs with supporting cells. In XY germ cells, novel findings include the predicted set of TFs regulating Bnc2, which is known to promote mitotic arrest, as well as the TFs POU6F1/2 and FOXK2 and their predicted targets that function in mitosis and signal transduction. In XX germ cells, the authors deconstruct the regulation of the premeiotic replication regulator Stra8, which reveals TFs involved in meiosis, retinoic acid signaling, pluripotency, and epigenetics among predictions; this finding, along with evidence supporting the regulatory potential of retinoic acid receptors in meiotic gene expression is an important addition to the debate over the necessity of retinoic acid in XX meiotic initiation. In addition, a self-regulatory network of other TFs is hypothesized in XX differentiating PGCs, including TFAP2c, TCF5, ZFX, MGA, and NR6A1, which is predicted to turn on meiotic and Wnt signaling targets. Finally, analysis of PGC-support cell interactions during sex differentiation reveals more interactions in XX, via WNTs and BMPs, as well as some new signaling pathways that predominate in XY PGCs including ephrins, CADM1, Desert Hedgehog, and matrix metalloproteases. This dataset will be an excellent resource for the community, motivating functional studies and serving as a discovery platform.Weaknesses:My one major concern is that the conclusion that PGC sex differentiation (as read out by transcription) involves chromatin priming is overstated. The evidence presented in the figures includes a select handful of genes including Porcn, Rimbp1, Stra8, and Bnc2 for which chromatin accessibility precedes expression. Given that the authors performed all of their comparisons between XX versus XY datasets at each timepoint, have they missed an important comparison that would be a more direct test of chromatin priming: between timepoints for each sex? Furthermore, it remains possible that common mechanisms of differentiation to XX and XY could be missing from this analysis that focused on sexspecific differences.

We thank the reviewer for their thoughtful assessment and suggestions, as stated here. We note that chromatin priming in PGCs prior to sex determination is a well-documented research finding (see references below), that is further supported by our single-nucleus multiomics data. To support these findings previously stated in the scientific literature, we included data demonstrating the asynchronous correlation between chromatin accessibility and gene expression during PGC sex determination. Specifically, we investigated the associations of differentially accessible chromatin peaks with differentially expressed gene expression for each PGC type (between sexes and across embryonic stages) using computational tools and methods that are well-established and applied by the research community. In our manuscript, we note that the patterns we identified support the potential role of chromatin priming in PGC sex determination. Nevertheless, we further highlight that a comprehensive profile of 3D chromatin structure and enhancer-promoter contacts in differentiating PGCs is needed to fully understand how changes to chromatin facilitate PGC sex determination.

References:

(1) Chen, M., et al. Integration of single-cell transcriptome and chromatin accessibility of early gonads development among goats, pigs, macaques, and humans. Cell Reports 41 (2022).

(2) Huang, T.-C. et al. Sex-specific chromatin remodelling safeguards transcription in germ cells. Nature 600, 737–742 (2021).

**Reviewer #3 (Public Review):**
Summary:Alexander et al. reported the gene-regulatory networks underpinning sex determination of murine primordial germ cells (PGCs) through single-nucleus multiomics, offering a detailed chromatin accessibility and gene expression map across three embryonic stages in both male (XY) and female (XX) mice. It highlights how regulatory element accessibility may precede gene expression, pointing to chromatin accessibility as a primer for lineage commitment before differentiation. Sexual dimorphism in these elements and gene expression increases over time, and the study maps transcription factors regulating sexually dimorphic genes in PGCs, identifying sex-specific enrichment in various transcription factors. Strengths:The study includes step-wise multiomic analysis with some computational approach to identify candidate TFs regulating XX and XY PGC gene expression, providing a detailed timeline of chromatin accessibility and gene expression during PGC development, which identifies previously unknown PGC subpopulations and offers a multimodal reference atlas of differentiating PGC clusters. Furthermore, the study maps a complex network of transcription factors associated with sex determination in PGCs, adding depth to our understanding of these processes.Weaknesses:While the multiomics approach is powerful, it primarily offers correlational insights between chromatin accessibility, gene expression, and transcription factor activity, without direct functional validation of identified regulatory networks.

As stated in our response above to a similar concern, we note that our research study represents a launching pad for which to identify future projects that will investigate candidates that may be involved in PGC sex determination, in further detail. With this rich dataset in hand, our goal in future research projects is to screen these candidates and discover their function in PGCs.

**Response to Recommendations**

**Reviewer #1 (Recommendations For The Authors):**
(1) Clarify at first introduction how combined ATAC-seq/RNA-seq mulitomics libraries were prepared, including if ATAC and RNA-seq data are from the same cell.

This information was added to the introduction of the revised manuscript.

(2) Clarify what the two technical replicates represent. Are they two libraries from the same gonad or the same pool of gonads? Are they from 2 different gonads?

The two independent technical replicates comprised different pools of paired gonads. This sentence was added to the methods section of the revised manuscript.

(3) In Supplemental Figure 1, there is substantial variation in the number of unique snATAC-seq fragments between some conditions. Could this create a systematic bias that affects clustering?

We recognize the concern that substantial variation in the number of unique snATAC-seq fragments between conditions could potentially create a systematic bias that affects clustering. However, we analyzed our snATAC-seq dataset with Signac, which performs term frequency-inverse document frequency (TF-IDF) normalization. This is a process that normalizes across cells to correct for differences in cellular sequencing depth. Given that sequencing depth was taken into account in our normalization and clustering procedures, and that the unbiased clustering of PGCs also reflects the sex and embryonic stage of PGCs, we are confident that the clustering of the snATAC-seq datasets closely reflects the biological variability present in the PGCs collected.

References:

Signac Website: https://stuartlab.org/signac/articles/pbmc_vignette

Stuart, T., Srivastava, A., Madad, S., Lareau, C. A., & Satija, R. (2021). Single-cell chromatin state analysis with Signac. Nature methods, 18(11), 1333-1341.

(4) In Figures 2a, 2e, 3a, and 3e, the visualization scheme is very difficult to follow. It's very hard to see the colors corresponding to average expression for many genes because the circles are so small. In addition, the yellow color is hard to see and makes it hard to estimate the size of the circle since the boundaries can be indistinct. I recommend using a different visualization scheme and/or set of size scales be used.

In Figures 2a, 2e, 3a, and 3e, we chose this color palette to be inclusive of viewers who are colorblind. The chosen colors are visible on both a computer screen and on printed paper. We also included a legend of the color scale and dot size representing the average expression and percent of cells expressing the gene, respectively. If the color cannot be seen, it is because the cell population is not expressing the gene.

(5) Perform in vivo validation (immunofluorescence or RNAscope) of at least some targets implicated in PGC development by this study.

Such validations (immunofluorescence staining of PORCN and TFAP2C) are now included in Figure 4 and the supplement.

(6) In line 351, the authors state that "we observed a strong demarcation between XX and XY PGCs at E12.5-E13.5." But in Figure 1j it looks like a reasonably high fraction of both XX and XY E12.5 cells are in cluster 1, which should mean that there is some overlap.

While it is true that Figure 1j shows overlap of both XX and XY E12.5 cells in cluster 1, we were commenting on the separation of E12.5 XX (clusters 4 and 5) and E12.5 XY (clusters 8 and 9) PGCs. We have modified the sentence beginning at line 351 to state that the separation between XX and XY PGCs occurs at E13.5.

(7) In lines 404-405: "We first linked snATAC-seq peaks to XY PGC functional genes". It is important to know how the peaks were linked to genes.

We added the following sentence to address this comment: “Peak-to-gene linkages were determined using Signac functionalities and were derived from the correlation between peak accessibility and the intensity of gene expression.”

(8) In Supplemental Figure 5c, the XX E11.5 condition has a substantially higher fraction of ATAC peaks at promoter regions compared to the others. Does this have statistical and biological significance?

This is an interesting observation beyond the scope of our manuscript. Many interesting questions arise from this study and it is our plan to investigate further in the future.

(9) Line 885: "The increased number of DA peaks at E13.5 may be the result of changes to chromatin structure as XX PGCs enter meiotic prophase I"; but in Figure 4b, there's only a modest increase in DAP number from E12.5 to E13.5 in XX PGCs, compared to a massive gain in XY PGCs.

In our manuscript, we comment on both phenomena: the doubling of differentially accessible peaks in XX PGCs from E12.5 to E13.5 and the massive increase in differentially accessible peaks in XY PGCs from E12.5 to E13.5. In our description of these results, we propose several hypotheses leading to these increases in differentially accessible peaks. As such, it cannot be ruled out that the changes to chromatin structure that occur during meiotic prophase I contribute to the gain in differentially accessible peaks in XX PGCs at E13.5, and we included this statement in the manuscript accordingly.

**Reviewer #2 (Recommendations For The Authors):**
(1) The methods state at line 141 that nuclei with mitochondrial reads of more than 25% were removed, however our understanding from the Bioconductor manual and companion manuscript (Amezquita, R.A., Lun, A.T.L., Becht, E. et al. Orchestrating single-cell analysis with Bioconductor. Nat Methods 17, 137-145 (2020). https://doi.org/10.1038/s41592-019-0654-x) is that snRNA-seq approaches remove mitochondrial transcripts entirely and datasets containing mitochondrial transcripts are thought to feature incompletely stripped nuclei. It is thought that mitochondrial transcripts participating in nuclear import may remain hanging on to the nuclear envelope and get encapsulated into GEMs. If the mitochondrial read cutoff of 25% was used intentionally to keep this potentially contaminating signal, please justify why this was done for this dataset.

We agree with the reviewer that the presence of mitochondrial transcripts may be potentially contaminating signal. In our preprocessing steps, we removed the mitochondrial genes and transcripts from our datasets so that they would not influence or affect our analyses. The following sentence was added to the methods section on snRNA-seq data processing: “Mitochondrial genes and transcripts were removed from the snRNA-seq datasets to eliminate any potentially contaminating signal.”

(2) Methods line 227: please include log2fold change and p-adjusted value cutoffs for GO enrichment.

We used clusterprofiler for our GO enrichment analysis. Our GO enrichment analysis did not include a log2fold change analysis and the p-adjusted value cutoff is stated in the methods.

(3) Results line 310: the claim that "At E12.5-E13.5, XY PGCs converged onto a single distinct population (cluster 7), indicating less transcriptional diversity among E12.5-E13.5 XY PGCs when compared to E12.5E13.5 XX PGCs (Fig1d)" would be strengthened if the authors quantified transcriptional distance with distance metrics such as euclidean or cosine distance.

We used a clustering approach to gain insights into the transcriptional diversity of PGC populations. Using an additional metric, such as Euclidean or cosine distance, would not provide meaningful information not already achieved by clustering or change the conclusions presented in the manuscript.

(4) Results line 317: the authors allude to Lars2 defining clusters 2 & 3 as a marker gene, but it is not clear why this is highlighted until the reader reaches the discussion, which alludes to the published role of Lars2 in reproduction. Please consider moving this sentence to the results section for clarity and perhaps expanding the discussion on the meaning.

To provide clarity, we added the statement “genes with reported roles in reproduction” to the results section.

(5) In Figure 2a, why do the authors choose to focus on Zkscan5 in XY PGCs when it is expressed by such a small portion of cells (<25%)? Do they assume that this is due to dropouts?

We chose to focus on Zkscan5 as an example because of its enriched and differential expression in male PGCs, the motif for Zkscan5 is not enriched in female PGCs, and the reported roles of Zkscan5 in regulating cellular proliferation and growth. Zkscan5 is an example of how candidate genes can be identified for further investigation.

(6) Line 461: "the population of E13.5 XX PGCs displaying the strongest Stra8 expression levels corresponded to the same population of XX PGCs with the highest module score of early meiotic prophase I genes (Figure 3c; Supplementary Fig. 3a-b)". However did the authors also consider examining the Stra8+ XX PGCs that do not robustly express meiotic genes to understand more about their differentiation potential?

We are thankful to the reviewer for this suggestion. However, this research question is beyond the scope of the manuscript. We plan to investigate further in future research studies.

(7) Line 505: "when we searched for the presence of RA receptor motifs in peaks linked to genes related to meiosis and female sex determination, we found that Stra8, Rec8, Rnf2, Sycp1, Sycp2, Ccnb3, and Zglp1 contain the RA receptor motifs in their regulatory sequences (Supplementary Figure 4g)." My read of the text is that the authors are not taking a side on the RA and meiosis controversy, but rather trying to reveal what the data can tell us, and the answer is that there is a strong signature linking RA to meiotic genes, which supports this as a valid biological pathway. But what is the strength of the RA>meiosis pathway compared to other mechanisms (which must be functioning in the triple receptor KO)? Perhaps the authors could take this analysis further with the following questions: (1) ask whether meiotic genes are more enriched in RA motifs compared to other expressed genes or other motifs (2) compare the strength of peak-gene correlations for all peaks containing RA receptor motifs vs. those with peaks for Zglp1, Rnf2, etc binding. The strengths of these correlations could provide clues to how much gene expression varies in response to RA exposure vs. modulation of these other factors and thus tell us something about how much RA is playing a role.

We agree with the reviewer that this is a very interesting and important question. We also thank the reviewer for their thoughtful suggestions on the types of bioinformatics analyses that could answer this question. However, the section on RA signaling during PGC sex determination is only a small part of the manuscript and would be better analyzed in greater detail in a future research study or publication.

(8) The shift from promoters in E11.5 XX PGCs to distal intergenic regions is fascinating. What can we learn about epigenetic reprogramming/methylation changes across gene bodies?

We agree with the reviewer that this is an interesting question about gene regulation in E11.5 XX PGCs. However, we prefer to analyze the epigenetic reprogramming changes across gene bodies in this cell population in additional research studies. Our purpose and goal for this section was to link differentially accessible chromatin peaks with differentially expressed genes to identify putative gene regulatory networks.

(9) Line 581: why did the authors choose to highlight and validate PORCN1 in PGCs? Please elaborate.

As stated in the manuscript, we chose to highlight and validate PORCN1 in PGCs because of its role in WNT signaling and because of the visibly strong correlation between chromatin accessibility at the XXenriched DAP in Fig. 4c (dashed box) and gene expression of PORCN1.

(10) Figure 5f would be easier to interpret if presented as two columns rather than a circle; show one line of the proteins and the other line with the transcripts so that each is on the same line and there are connections between them.

This comment is related to stylistic preferences. The purpose of Fig. 5f is to demonstrate that the candidate transcription factors may regulate the expression of other enriched transcription factors. Figure 5f figure accomplishes this goal.

(11) Line 640: "The predicted target genes of TCFL5 totaled 74% (367/494) of all DEGs with peak-to-gene linkages in XX PGCs". This seems like a high number and a lot of work for just TCFL5; given the overlap between other TFs and target genes, how many of these 367 target genes overlap with other TFs?

We agree with the reviewer that this is an important declaration to make. We added the following sentence to the results section on TCFL5: “A large majority of the predicted target genes of TCFL5 were also predicted to be the target genes of the enriched TFs presented in Fig. 5e, e.g., the predicted target genes of these TFs overlapped with 4%-100% of the predicted target genes of TCFL5.”

(12) The presentation of TCFL5 in the results section would make more sense with the additional mention of reproductive phenotypes already known (currently in the discussion Lines 914-917). I would furthermore suggest that the discussion goes into more depth on the difference between the regulatory network of TCFL5 in XX meiosis vs XY.

We thank the reviewer for this comment, however, we already state in the results section that TCFL5 is known to influence XX PGC sex determination.

(13) In the Methods, please state more clearly for those not familiar that the genetic background of mice is mixed.

We described the mice with their official names, which provides the context of their genetic backgrounds.

(14) Please specify which morphologic criteria were used to verify the stage of embryos in the methods.

We added the following text to the methods section of the revised manuscript: “Plug date was used to determine the stage of embryos collected for single-nucleus RNA-seq and ATAC-seq. The stage of E11.5 embryos was confirmed by counting somites. The stage of embryos collected at E12.5 was confirmed by the morphological presence of the vessel and cords of the testes collected from XY embryos. Similarly, we confirmed the stage of embryos collected at E13.5 by the size of the gonads, the presence of more distinct cords in the testes of XY embryos, and the elongation of the ovaries of XX embryos.”

(15) The total number of cells and PGCs that passed QC and are included in UMAPS should be stated.

The requested information was added to the legend for Fig. 1 of the revised manuscript: “The number of PGCs per sex and embryonic stage are: 375 E11.5 XX PGCs; 1,106 E12.5 XX PGCs; 750 E13.5 XX PGCs; 110 E11.5 XY PGCs; 465 E12.5 XY PGCs; and 348 E13.5 XY PGCs.”

(16) The order of timepoints changes between figures, and this is not for any obvious reason. Please make it consistent. Figures 1 and 6 list XX 11.5, 12.5, 13.5, and the same for XY, but Figures 2, 3, and 4 use the reverse order: XY E13.5, E12.5, E11.5, and then XX.

We thank the reviewer for this comment. However, we chose this order for each of the figures to match the coordinates of the graphs and where we would expect the reader to begin reading the graph first. For example, in Figure 3a, XX E11.5 is closest to the x-axis and would be expected to be read first.

(17) In Figure S2 the colors of clusters are hard to distinguish, and it is suggested that the cluster numbers should be listed above each colored bar to avoid frustration.

We made the suggested correction to Figure S2.

(18) In Figures 2e and 3e: what do the dashed boxes indicate?

The dashed boxes are to guide the reader’s eyes to the fact that the order of transcription factors/genes under the Cistrome DB regulatory potential score and gene expression plots are the same.

(19) In Figure 5a: break panels into i-iv so that the in-text call-outs are not all the same.

We made the suggested correction to Figure 5a and modified the in-text call-outs.

(20) Please indicate XX in Figure 5e and XY in Figure 5l.

We made the suggested correction to Figure 5e and 5l.

(21) In Figure S5c: Please reorganize DA chromatin peak charts so that columns are XX and XY with rows at the same timepoint.

We made the suggested correction to Figure S5c.

(22) In Figure S7a: please make images larger so that the overlapping expression of PORCN and TRA98 is more visible, and consider adding a more magnified panel.

This image is now included in the main text, with expanded panels.

(23) Line 742-754: this seems like a long introduction for the results section; please consider tightening it up.

We believe this text is important and necessary to provide context to the bioinformatics analyses of cell signaling pathways in PGCs. Not all readers will be familiar with the ligand-receptor signals between gonadal support cells and PGCs, and this text provides details on which signaling pathways are known to direct sex determination of PGCs.

(24) For UMAP plots in Figures 2c, 3c, S3b, and S4b, the text overlaid with the timepoints and sexes onto the UMAP plots is misleading, as it allows the reader to presume that the entire group of cells for a given sex/timepoint is located in the location of the text overlay. However, from the UMAP plots in Figure 1i-j, it is clear that the cells from a given sex/timepoint are actually spread across multiple identified clusters. Thus, the overlaid text obscures the important heterogeneity detected. To better represent the actual locations on the UMAP plot of cells from each sex/timepoint, it would be better to show inset density plots alongside these UMAP plots so the reader can locate the cells for themselves.

We thank the reviewer for this comment. However, we chose this formatting to offer simplicity and ease of understanding to our UMAPs in addition to highlighting the general biological patterns of gene expression. If the reader is interested in discerning more of the heterogeneity of the UMAPs, they may refer back to Figure 1.

**Reviewer #3 (recommendations for the authors):**
There are some errors or places that need clarification or corrections:(1) Figure 1f, according to the graph, it should be 8 clusters, not 9.

There are 9 clusters because the numbering for the clusters start at ‘0’.

(2) Why did cluster 8 have so many different states of cells from both sexes?

The identification of cluster 8 is likely an artifact of sequencing, and would require several different analyses to figure out why cluster 8 has many different states of cells from both sexes. While this will address a technical issue associated with the dataset, this will not change any major conclusions of the study.

(3) Figure 1i, shouldn't that be ten instead of eleven?

There are 11 clusters because the numbering for the clusters start at ‘0’.

(4) Figure 2a, zkscan expression level comparison was not so obvious as the bubble size was small. How many folds of differences from xx pgc?

There is a 1.5 fold increase in the expression of Zkscan5 between XY and XX PGCs at E13.5. We included this information in the revised manuscript.